# A Reoptimization Framework for Mixed Integer Linear Programming with Dynamic Parameters

## Abstract

Many real-world applications, such as logistics, routing, scheduling, and production planning, involve dynamic systems that require continuous updates to solutions for new Mixed Integer Linear Programming (MILP) problems. These environments often require rapid responses to slight changes in parameters, with time-critical demands for solutions. While reoptimization techniques have been explored for Linear Programming (LP) and specific MILP problems, their effectiveness in general MILP is limited. In this work, we propose a two-stage reoptimization framework for efficiently identifying high-quality feasible solutions. Specifically, we first utilize the historical solving process information to predict the high confidence solving space for modified MILPs to contain high-quality solutions. Based on the prediction results, we fix a part of variables to apply the prediction intervals and use the Thompson Sampling algorithm to determine the set of variables to fix by updating the Beta distributions based on solutions obtained from the solver. Extensive experiments across nine reoptimization datasets show that our VP-OR outperforms the state-of-the-art methods, achieving higher-quality feasible solutions under strict time limits and demonstrating faster convergence with smaller primal gaps in the early stages of solving.

## 1 Introduction

Traditional combinatorial optimization problems require finding solutions for a single instance. However, many real-world scenarios, such as system control (Marcucci & Tedrake, 2020), railway scheduling (Zhang et al., 2020) and production planning (Dunke & Nickel, 2023; Cedillo-Robles et al., 2020), involve systems that change dynamically over time. Thus, throughout the continuous operation of such systems, it is required to compute solutions for new Mixed Integer Linear Programming (MILP) problems, which are similar to the previous instances but differ in some parameters in specific model elements such as objective functions, constraints, and variable bounds. Traditionally, each of these new MILP instances is solved from scratch, which overlooks the opportunity to leverage valuable information from the previously solved instances. This can be computationally expensive on resource and it is usually a challenging task to make a high-quality operation plan in a short period for time-critical applications.

Reoptimization techniques have been well-studied for the LP case (John & Yıldırım, 2008) and heuristic algorithms for some special MILP problems, e.g., the railway planning problem (Blair, 1998), general assignment problems (NAUSS, 1974) and other combinatorial problems (Libura, 1996; 1991; Sotskov et al., 1995). However, the functionality of these techniques for general MILPs is very limited. The earliest reoptimization methods Ralphs & Güzelsoy (2006); Ralphs et al. (2010) were primarily based on duality theory and focused on sequences of MIPs where only the right-hand side changes. These approaches leveraged dual information obtained through primal algorithms to enable "warm starting", accelerating the resolution of subsequent problems. Later research (Gamrath et al., 2015) extended these methods to broader scenarios, incorporating techniques like reusing branch-and-bound trees. In such methods, the modified problem is treated as a subproblem of the base problem, or if only the objective function changes, the search can "continued" from the last known search boundary. Specifically, these methods use the leaf nodes of the base problem's branch-and-bound tree as starting points for solving the modified problem. Build-

ing on this, more recent work (Patel, 2024) addressed even more complex reoptimization scenarios, where, apart from the number of variables and constraints, all other parameters can undergo upward or downward perturbations, including the objective function, variable bounds, matrix coefficients, and constraint right-hand side values. Their approach centers on reoptimization methods built on the SCIP solver (Bestuzheva et al., 2021), where a series of past solutions is preserved, allowing the method to assess whether portions of these solutions can be reused for the new problem. They also explore reusing branching strategies and adjust parameters related to the invocation of cutting planes and heuristic algorithms, fine-tuning the solver's behavior to better tackle the modified problem. The limitations of their approach are twofold. Firstly, the optimal solution from the original problem may no longer be valid for the new problem. This is because the range of variable values for the optimal solution can shift significantly, even with small modifications (Guzelsoy, 2009). Secondly, reusing branching strategies and adjusting parameters mainly saves time on selecting variables, generating heuristics and cutting planes but does not reduce the overall size or complexity of the problem.

Inspired by recent work on end-to-end problem solving (Han et al., 2023; Khalil et al., 2022; Ye et al., 2023; 2024; Nair et al., 2020), we aim to leverage GNNs to predict how optimal solutions change when MIP parameters vary. However, reoptimization presents unique challenges: Firstly, in many real-world scenarios that rely on reoptimization techniques, integer and continuous variables are common and natural representations. For example, production quantities in manufacturing are integers (Cedillo-Robles et al., 2020), while power levels in energy optimization problems are continuous variables (Yokoyama et al., 2002). However, most existing end-to-end machine learning-based methods primarily focus on predicting solutions for binary variables (Han et al., 2023; Khalil et al., 2022) and only use the optimal solution from the previous problem without leveraging the intermediate solving process. Secondly, In reoptimization scenarios, there is a pressing real-world need for quickly obtaining high-quality feasible solutions (Marcucci & Tedrake, 2020; Zhang et al., 2020). While current end-to-end methods handle inaccuracies in variable predictions using techniques like Large Neighborhood Search (LNS) (Han et al., 2023; Ye et al., 2024), which explores solutions near the predicted values. Although LNS intuitively narrows the search space by focusing on the neighborhood of predicted values, the pure LNS method without the fix strategy does not actually decrease the problem's variable size. In fact, if the search range is not well-tuned, the added constraints can increase the complexity of the problem, leading to additional computational overhead (Carchrae & Beck, 2009).

In this paper, we propose a two-stage reoptimization framework designed to solve near-optimal solutions for modified large-scale MILP instances in dynamic parameter scenarios. The framework consists of a **V**ariable **P**rediction model and an **O**nline **R**efinement module (VP-OR). The variable prediction model employs a Graph Neural Network (GNN) to analyze changes in problem structure and historical branch-and-bound processes. It predicts a marginal probability of each binary variable and the feasible ranges of integer and continuous variables. The online refinement module utilize Thompson Sampling to iteratively select the variable to apply the prediction interval, gradually improving the overall solution quality. The overall framework is outlined in Figure 1.

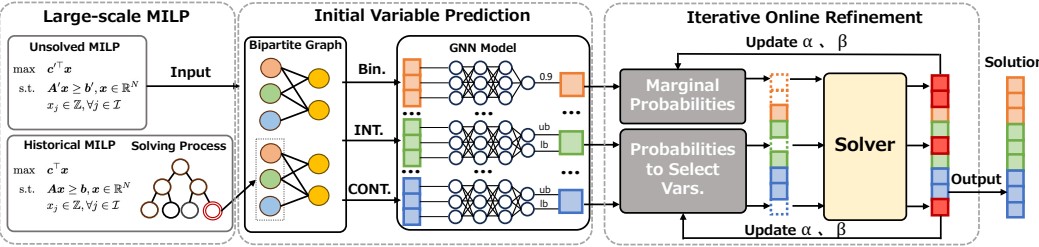

Figure 1: Illustration of our proposed two-stage reoptimization framework. Our approach first predicts a marginal probability of each binary variable and the feasible ranges of integer and continuous variables utilizing a graph neural network (GNN), and then employs the Thompson Sampling algorithm to iteratively select the variable to apply the interval to solved for near optimal solutions.

We compare VP-OR against the leading reoptimization method (Patel, 2024), two end-to-end machine learning-based baselines (Nair et al., 2020; Han et al., 2023), and the open-source solver SCIP (Bestuzheva et al., 2021) across nine reoptimization datasets. The results indicate that VP-OR outperforms the other methods in delivering highly accurate solutions under strict time limits. In

addition, we evaluate the performance over a longer duration, revealing that VP-OR converges more rapidly, achieving smaller primal gaps compared to the other methods.

## 2 PRELIMINARIES

### 2.1 MIXED INTEGER LINEAR PROGRAMMING(MILP)

A general MILP problem is defined by a set of decision variables, where a subset or all variables are required to be integers. For simplicity, we assume that the objective of our MILP problems is to seek the minimum value, and a MILP instance can be formulated as formulated below:

$$\begin{align} \min \quad & \boldsymbol{c}^\top \boldsymbol{x} \\ \text{s.t.} \quad & \boldsymbol{A}\boldsymbol{x} \geq \boldsymbol{b}, \boldsymbol{x} \in \{0,1\}^p \times \mathbb{Z}^q \times \mathbb{R}^{n-p-q} \\ & \boldsymbol{l} \leq \boldsymbol{x} \leq \boldsymbol{u} \end{align} \tag{1}$$

Here, $\boldsymbol{x}$ represents the $n$ decision variables, where $\boldsymbol{c}, \boldsymbol{l}, \boldsymbol{u} \in \mathbb{R}^n$ are the objective coefficients, and the lower and upper bounds, respectively. The matrix $\boldsymbol{A} \in \mathbb{R}^{m \times n}$ is the coefficient matrix of the constraints, and $\boldsymbol{b} \in \mathbb{R}^m$ is the right-hand side vector.

### 2.2 MODIFIED MILP PROBLEM

We consider scenarios similar to those described by Patel (2024), involving a series of MILP instances based on an MILP (base instance) taken from a specific application. Each subsequent instance (modified instance) is modified from the previous one with random perturbations and rotations to parameters such as the objective vector, constraints, and variable bounds. The previous instances has been solved to optimality. They provide not only the optimal solution but also detailed records of intermediate computational steps, such as selected branches and basis variables at each node's LP relaxation. These records can be strategically leveraged in the reoptimization algorithm to accelerate the solving process for the modified instances.

### 2.3 BIPARTITE GRAPH FOR MILP

An MILP problem can be effectively represented as a weighted bipartite graph $G = (V \cup C, E)$ (Nair et al., 2020; Gasse et al., 2019). Each vertex in $V$ corresponds to a variable of the MILP, and each vertex in $C$ represents a constraint. An edge $(v_i, c_j)$ connects a variable vertex $v_i$ with a constraint vertex $c_j$ if the variable is involved in the constraint. The edge set $E \in \mathbb{R}^{m \times n \times e}$ represents the edge features, where $m$ and $n$ denote the number of constraints and variables, respectively, and $e$ indicates the dimension of the edge attributes.

### 2.4 ONLINE CONTEXTUAL THOMPSON SAMPLING

Thompson Sampling is a heuristic strategy used in decision-making scenarios like the multi-armed bandit (MAB) problem (Zhao, 2022). This method is used for choosing actions according to their expected rewards, which are continuously updated using Beta probability distributions $Beta(\alpha, \beta)$ (Gupta & Nadarajah, 2004). The Beta distribution forms a family of continuous probability distributions over the interval $(0, 1)$. The probability density function (pdf) of a $Beta(\alpha, \beta)$ distribution, where $\alpha > 0$ and $\beta > 0$, is given by: $f(x; \alpha, \beta) = \frac{\Gamma(\alpha+\beta)}{\Gamma(\alpha)\Gamma(\beta)} x^{\alpha-1}(1-x)^{\beta-1}$, where $\Gamma(\cdot)$ is the Gamma function. The mean of the $Beta(\alpha, \beta)$ distribution is $\frac{\alpha}{\alpha+\beta}$, and as the parameters $\alpha$ and $\beta$ increase, the distribution becomes more concentrated around the mean. The beta distribution is useful for Bernoulli rewards because if the prior is a $Beta(\alpha, \beta)$ distribution, then after observing a Bernoulli trial, the posterior distribution is simply $Beta(\alpha + 1, \beta)$ or $Beta(\alpha, \beta + 1)$, depending on whether the trial resulted in a success or failure, respectively.

## 3 INITIAL VARIABLE PREDICTION

In this section, our goal is to train a GNN model to predict a feasible interval containing the modified problem's optimal solution by utilizing the subproblem containing the optimal solution of the base problem and analyzing how the parameters of the MILP change.

### 3.1 GRAPH REPRESENTATION

The feature extraction process is divided into two parts: the base instance and the modified instance. For the modified instance, we represent it using a classic bipartite graph structure (Gasse et al.,

2019). For the base instance, we aim to extract additional historical solving information to predict how the optimal solution may change under small perturbations in the MILP.

In MILP, integer variables are often relaxed to continuous values to apply duality concepts. However, the dual problem from the relaxed problem may not directly reflect the relationship between the optimal solution and constraints under integer restrictions. We address this challenge by leveraging a key property of branch-and-bound trees: the final leaf node that yields the MILP optimal solution has the characteristic that its LP relaxation solution is also an integer solution. The leaf node represents a subproblem of the original MILP, distinguished by the addition of a series of branching constraints. We include the feasible basic variables and dual solutions of the leaf node as features, which are commonly applied in LP sensitivity analysis (Higle & Wallace, 2003), aiming to capture which variables and constraints are sensitive to parameter changes. This approach significantly improves the accuracy of binary variable predictions compared to traditional end-to-end solving methods, which rely solely on modeling the problem as a bipartite graph and optimal solution values. In Appendix C.5, we present the comparison results between the reuse of historical solving information and the traditional vanilla bipartite graph predictions. A list of the features used in our graph representation is detailed in Table 5 in Appendix A.

### 3.2 GNN-Based Initial Variable Prediction

Classic end-to-end approaches (Khalil et al., 2022; Han et al., 2023) are specifically designed for binary variables and predict a n-dimension vector $(p_\theta(x_1 = 1; M), \ldots, p_\theta(x_p = 1; M))$ to represent the conditional probability of $p$ binary variables. However, these methods can not work well in many real-world scenarios, which mainly contain integer and continuous variables. For instance, in the dataset named "vary_matrix_rhs_bounds" in the MIP Workshop 2023 Computational Competition (Bolusani et al., 2023), there are 27,710 variables but only 400 binary variables. Therefore, VP-OR proposes a confidence threshold method specifically designed for integer and continuous variables. Specifically, we represent these values with binary bits and hope to predict the conditional probability of each bit, however there a new challenge arises: representing variable values through high-dimensional binary bits is computationally prohibitive (Nair et al., 2020). A common technique is to decompose the high-dimensional distribution into lower-dimensional ones. This is suitable for our problem because we only focus on the prediction interval to reduce problem scales but do not need to predict an accurate value for continuous and integer variables.

To reduce the dimensionality of integer variables, we apply a logarithmic transformation before converting the integer values into binary representations. In this process, the integer values can potentially be negative. While two's complement is typically used to represent negative numbers in binary form (Baugh & Wooley, 1973), it is less intuitive for tasks that involve magnitude interpretation, such as logarithmic transformations. Instead, we introduce a sign bit $s \in \{0, 1\}$ to separately capture the sign of the variable, making the magnitude and sign easier to handle. Specifically, we record the optimal value $v_i$ of the variable $x_i$ in the base instance, we calculate its logarithmic scale and binary sign bit $s$ as follows:

$$\mathbf{b}(v_i) = \text{bin}\left(\lfloor \log_2(|v_i| + 1) \rfloor\right), \quad s = \begin{cases} 0 & \text{if } v_i \leq 10^{-5}, \\ 1 & \text{otherwise.} \end{cases}$$

where the vector $\mathbf{b}(v_i)$ represents the binary representation of the logarithmic value of $v_i$, prefixed by the sign bit $s$, and $\text{bin}(\cdot)$ denotes the binary conversion of the logarithmic value.

Based on the predicted initial variable vector $\tilde{\mathbf{b}}(v_i)$, we apply the confidence threshold method (Yoon, 2022) to filter the predicted probabilities and distinguish between confident and uncertain predictions. For the binary digits with high confidence, the binary digits are fixed to their predicted values. For uncertain binary digits (i.e., those with probabilities between $0.1$ and $0.9$), we allow them to vary between $0$ and $1$. Specifically, we establish the upper and lower bounds of the predicted binary encoding $\tilde{\mathbf{b}}(v_i)$ by setting the uncertain binary digit to its maximum value $1$ for the upper bound and its minimum value $0$ for the lower bound.

To further determine the variables' feasible range, the upper and lower bound are converted back into their corresponding integer forms, denoted as $k_{ub}$ and $k_{lb}$. For positive variables ($s = 1$), we represent the variable's value of the optimal solution in the form $2^k + m$, where $k \geq 0$, $0 \leq m \leq 2^k - 1$. From the inequality $k < \lfloor \log_2(|v_i| + 1) \rfloor \leq k + 1$, the predicted range for the variable

Table 1: Comparison of variable prediction accuracy for different datasets. This table presents the number of variables and mispredicted variables across different types (binary, integer, and continuous) when using GNN-based predictions. Mispredicted variables represent those whose predicted bounds or values differ from the optimal solution.

| Var. num. | bnd_1 | mat_1 | obj_1 | obj_2 | rhs_1 | rhs_2 |
|---|---|---|---|---|---|---|
| binary var. | 2993.0 | 500.0 | 360.0 | 355.0 | 12510.0 | 500.0 |
| mispredicted binary var. | 8.2 | 37.4 | 5.6 | 0.2 | 64.3 | 0.0 |
| integer var. | 124.0 | 0.0 | 0.0 | 150.0 | 0.0 | 0.0 |
| mispredicted integer var. | 17.4 | 0.0 | 0.0 | 5.0 | 0.0 | 0.0 |
| continuous var. | 0.0 | 302 | 0.0 | 240.0 | 250.0 | 500.0 |
| mispredicted continuous var. | 0.0 | 0.0 | 0.0 | 16.8 | 0.0 | 1.6 |

lies between $2^{k_{lb}} - 1$ and $2^{k_{ub}+1}$. For negative variables ($s = 0$), the ranges are symmetrically calculated, spanning from $-2^{k_{ub}+1}$ to $-2^{k_{lb}} + 1$. For continuous variables, we first round them to the nearest integer and then process them similarly to integer variables.

## 4 ITERATIVE ONLINE REFINEMENT

Due to the potential distance between the initial predicted solution and the optimal solution, the initial prediction of variable confidence probabilities may be biased. However, by leveraging the feasible range predictions for variables, as described in Sec. 3, we can significantly reduce the problem's variable scale and search space, thereby substantially lowering the computation time. This reduction creates the opportunity for iterative solution refinement.

In this section, we first introduce our observation that only a very small number of variable predictions are inaccurate. However, identifying these inaccurately predicted variables is challenging due to their presence within a large variable space. To address this, we employ the Thompson Sampling algorithm to refine the solving space by selecting the predicted variable ranges to apply and adjusting the marginal probabilities of the binary variables. Based on the results from each iteration, we update the fixed variables for the next round of optimization, ensuring that the search focuses on the most promising regions of the solution space.

### 4.1 OBSERVATION

We aim to understand the prediction accuracy of the binary, integer and continuous variables. We test the prediction accuracy of GNN-based models on a variety of datasets, which were carefully selected to represent different types of parameters, including variable bounds, objective function coefficients, matrix parameters, and right-hand side constraints. Table 1 provides the number of variables and the mispredicted variables for each dataset.

From the results, we observe that the inaccuracies in predicted variable ranges and values are typically concentrated in a small subset of the variables. This is particularly evident in the integer and continuous variables, where only a few ranges deviate from the true feasible regions. Similarly, for binary variables, the majority of predictions are accurate, with only a limited number of cases where the predicted value differs from the optimal solution. However, these inaccurately predicted variables can still significantly affect the solution quality if not properly identified and addressed.

With this observation, it is reasonable to accelerate the solving process for MILP problems by fixing variables in the partial solution. To simplify the formulation, we denote the constraint space of the modified instance as: $S = \{x \in \{0,1\}^p \times \mathbb{Z}^q \times \mathbb{R}^{n-p-q} : (A + \Delta A)x \geq (b + \Delta b), (l + \Delta l) \leq x \leq (u + \Delta u)\}$. Specifically, the sub-problem of an instance using the fixing strategy with the predicted binary value $\tilde{x}$, the predicted lower bound $\tilde{l}$ and upper bound $\tilde{u}$ can be formulated as:

$$\min_{x \in S(\tilde{x}, I) \cap S} (c + \Delta c)^\top x \tag{2}$$

where the learning-based constraint set $S(\tilde{x}, I)$ is defined as: $S(\tilde{x}, I) = \{x \in \{0,1\}^p \times \mathcal{Z}^q \times \mathcal{R}^{n-p-q} : x_i = \tilde{x}_i, i \in \{1, 2, \ldots, p\} \cap I, \tilde{l}_j <= x_j <= \tilde{u}_j, j \in \{p+1, p+2, \ldots, n\} \cap I\}$, and $\tilde{x}_i$ represents the predicted probability for binary variables $x_i$, and $\tilde{l}_j$ and $\tilde{u}_j$ are the predicted lower and

upper bounds for integer or continuous variables $x_j$. Here, $I$ is a subset of $\{1, 2, \ldots, n\}$, representing the set of selected related variables in the constraint set. However, for $i \in I$, if $\tilde{x}_i \neq x_i^*$ or if $\tilde{l}_j > x_j^*$ or $\tilde{u}_j < x_j^*$, where $x_i^*$ denotes the optimal value of the variable $x_i$ in the modified problem, the fixing strategy may lead to suboptimal solutions or even infeasible sub-problems. Identifying the appropriate set $I$ to avoid these inaccurate predictions is challenging, particularly when handling large-scale problems where the search space is vast, and the number of variables is substantial.

Interestingly, through a number of experimental tests, as shown in the Table 2, we found that when fixing a portion of the variables, the solution time of the problem can become very short. For comparison, we included the solving times of SCIP without reoptimization and those obtained by large neighborhood search (LNS) methods. In cases where incorrect variable fixing caused infeasibility, we randomly select variables multiple times and compute the average time taken to find feasible solutions across these selections. The results show the average solving time across all instances within each dataset. When we add the estimated integer and continuous variables, we can increase the problem solving efficiency to **3-10 times** by fixing only the binary variables, which motivates us to choose more accurate variables based on the feedback of each solution, and gradually update the initial values of predictions based on the solution values of the binary variables.

Table 2: Comparison of solving times under different percentages of fixed variables (50% and 70%) for binary and all variable types.

| Solving time | bnd_s1 | mat_s1 | obj_s1 | obj_s2 | rhs_s1 | rhs_s2 |
|---|---|---|---|---|---|---|
| SCIP original solving time | 356.09 | 541.78 | 570.69 | 200.10 | 546.40 | 68.67 |
| LNS (50% binary variables) | 328.50 | 111.54 | 123.71 | 306.21 | 287.91 | 59.85 |
| LNS (70% binary variables) | 335.51 | 497.24 | 703.86 | 307.78 | 247.45 | 81.27 |
| Fix 50% variables (only binary) | 17.61 | 0.60 | 42.69 | 5.71 | 9.99 | 29.00 |
| Fix 70% variables (only binary) | 4.92 | 0.38 | 2.18 | 3.08 | 5.42 | 13.78 |
| Fix 50% variables (all) | 3.87 | 0.57 | 54.96 | 0.24 | 9.62 | 3.97 |
| Fix 70% variables (all) | 0.71 | 0.37 | 2.43 | 0.23 | 4.83 | 3.53 |

## 4.2 Online Variable Fixing Strategy

---
**Algorithm 1** Overall Thompson Sampling Framework

---
1: **Input:**
2: Predicted marginal probabilities $p_i$ for binary variables $x_i$,
3: Predicted bounds for continuous/integer variables $x_j$.
4: **Initialize prior distributions**:
5: $\alpha_i, \beta_i \sim \text{Beta}(p_i + 10^{-5}, 1 - p_i + 10^{-5})$ for binary variables,
6: $\alpha_j, \beta_j \sim \text{Beta}(1, 1)$ for continuous/integer variables.
7: **for** each time step $t = 1, 2, \ldots$ **do**
8:      Sample $\mu_i \sim \text{Beta}(\alpha_i, \beta_i)$ for all binary variables $x_i$
9:      Sample $\mu_j \sim \text{Beta}(\alpha_j, \beta_j)$ for all continuous/integer variables $x_j$
10:      **Select Variables:**
11:      Rank and select the top $a\%$ for binary and continuous/integer variables
12:      **Fix Selected Variables:**
13:      For binary variables, fix values using Bernoulli distribution with probability $\mu_i$
14:      For continuous/integer variables, apply predicted bounds
15:      Solve subproblem with selected variable values to obtain solution $x_t$ and objective $z_t$
16:      **Update Parameters:**
17:      **if** $z_t$ is better than the best objective value $z^*$ from previous iterations **then**
18:          Update values of $\alpha$, $\beta$ for variables (detailed in Algorithm 2 in Appendix B)
19:          $z^* \leftarrow z_t$
20:      **end if**
21:      **if** solution becomes infeasible **then**
22:          Apply relaxation mechanism (detailed in Algorithm 3 in Appendix B)
23:      **end if**
24: **end for**

---

In scenarios where the prediction results of certain variables may be inaccurate, existing methods primarily pre-screen binary variables using metrics such as model-predicted probability scores (Han et al., 2023; Khalil et al., 2022). However, as illustrated in Table 1, relying solely on machine learning predictions is unreliable, as even high-probability binary variables can be misclassified. This misclassification can degrade the quality of the solution or, in some cases, result in an unsolvable problem. Furthermore, while integer and continuous variables are represented in binary form, averaging the confidence scores across these binary representations is not a meaningful criterion for selection. This is because we are not concerned with the accuracy of individual bits within the binary encoding. Instead, our primary objective is to ensure that the predicted range encompasses the optimal solution.

**Problem Statement.** To alleviate these issues, we model the problem as a stochastic multi-armed bandit (MAB) problem, where an algorithm must decide which arm to play at each time step $t$, based on the outcomes of the previous $t - 1$ plays. For binary variables, we select a fixed set of $x\% \times p$ binary variables, where $p$ represents the total number of binary variables, and $a\%$ is a predetermined probability threshold. For each binary variable in this selected set, we fix its value to either 0 or 1. As a result, we have $2^{C_p^{a\% \times p}}$ possible arms, where $C_p^{x\% \times p}$ denotes the combination (binomial coefficient) $\binom{p}{x\% \times p}$. For integer and continuous variables, the setup is similar, but instead of selecting variable values directly, we focus on whether or not to apply upper and lower bound constraints based on predictions. We select a fixed set of $x\% \times (n - p)$ variables, where $(n - p)$ represents the number of integer and continuous variables, and $a\%$ remains the predetermined threshold. This yields $C_{n-p}^{x\% \times (n-p)}$ possible arms. At each time step $t = 1, 2, 3, \ldots$, one of the $N$ arms must be chosen. After selecting an arm, the values of the corresponding variable set are fixed to their predicted values, thereby reducing the original problem to a subproblem. We then solve this subproblem and obtain a reward $r_{a_t} \in \{0, 1\}$. If the solution obtained in this iteration improves upon all previous solutions, we set $r_{a_t} = 1$; otherwise, $r_{a_t} = 0$. The objective is to discover better solutions with as few iterations as possible, approaching the optimal solution efficiently. This aligns with the typical MAB goal of maximizing the expected total reward over a time horizon T, i.e., $\mathbb{E}\left[\sum_{t=1}^{T} r_{a_t}\right]$, where $a_t$ represents the arm played at time $t$, and the expectation is taken over the random choices of $a_t$ made by the algorithm.

We do not use the seemingly more intuitive approach of directly using the solution's objective value as the reward, as the reward could cause the model to favor solutions that are "good but not optimal", reducing the motivation for exploration. By rewarding only when the current solution is better than all previous ones, we can more clearly distinguish which arms lead to true improvements. Additionally, aiming for the best possible objective value in every round is unnecessary, as our main concern is the overall speed of convergence.

We base our approach on a simplifying assumption commonly used in prior work (Nair et al., 2020; Han et al., 2023) that treats each variable as independent of others. This assumption enables us to update the values of $\alpha$ and $\beta$ separately for each variable. Inspired by Patel (2024), in the initial step, we provided the solutions from previous instances as hints for the "completesol" heuristic method during the presolve phase, effectively using the base solution as a warm start.

**Update $\alpha$, $\beta$ for binary variables.** For binary variables $x_i, i \in \{1, 2, \ldots, p\}$, we initialize the prior distribution $\text{Beta}(p_i + 10^{-5}, 1 - p_i + 10^{-5})$, where $p_i$ represents the marginal predicted probability of the variable being fixed to 1. The value $\mu_i$, sampled from the Beta distribution, represents the probability that fixing binary variable $x_i$ to 1 will lead to a better solution (i.e., obtaining $r = 1$). We rank variables based on $\min(\mu_i, 1 - \mu_i)$, selecting the lowest $a\%$, and sample fixed values from the Bernoulli distribution with probability $\mu_i$. At each iteration $t$, the priors for unselected binary variables are updated based on their observed outcomes: For unselected binary variables, we set $\alpha_i = \alpha_i + 1$ when $x_i = 1$, and set $\beta_i = \beta_i + 1$ when $x_i = 0$. For selected binary variables, if the current solution $x_t$ is better than the previous best $x_{t-1}^*$, compare the set of selected variables $a_t$ with the previously best set $a_{t-1}^*$. For variables where the current value is 1 but was 0 in $a_{t-1}^*$ (or was not selected), we set $\alpha_i = \alpha_i + 1$. If the current value is 0, and it was 1 in $a_{t-1}^*$ (or was not selected), we set $\beta_i = \beta_i + 1$.

**Update $\alpha$, $\beta$ for integer and continuous variables.** For integer and continuous variables, we initialize the prior distribution as $\text{Beta}(1, 1)$, representing a uniform prior over the probability space.

The sampled value $\mu_j$ indicates the likelihood that imposing predicted bounds on variable $x_j$ will lead to a better solution. We select the top $a\%$ of these variables based on their $\mu_j$ values and apply the predicted upper and lower bounds. The priors for continuous/integer variables are updated as follows: For unselected variables, if the variable's actual value in the solution falls within the predicted bounds, we set $\alpha_j = \alpha_j + 1$. If it does not satisfy the predicted bounds, we set $\beta_j = \beta_j + 1$. For selected variables, when the current solution $x_t$ is better than the previous best $x_{t-1}^*$ (i.e., $r_t = 1$), we compare the set of selected variables $a_t$ and the previously best set $a_{t-1}^*$. If $x_j$ was unselected in $a_t$ but was selected in $a_{t-1}^*$, we set $\beta_j = \beta_j + 1$. In contrast, if a variable $x_j$ was selected in $a_t$ but not in $a_{t-1}^*$, no immediate conclusion about its benefit can be drawn, since the objective is to rule out incorrect bound predictions.

In our algorithm, we aim to avoid being trapped in local optima. To encourage exploration, no penalty is given when the current solution performs worse than previous iterations. If the current solution becomes infeasible, this is often due to incorrect predictions on the fixed variables. In such cases, a relaxation mechanism is triggered. The variables that have been fixed are divided into 10 groups. We iteratively solve 10 subproblems, where in each subproblem, one group of variables is relaxed (i.e., their bounds are loosened). This process continues until a feasible solution is found by adjusting the bounds of variables. In our testing process, this relaxation is handled sequentially to ensure fairness and accuracy. However, under conditions allowing parallelization, each group could be processed concurrently to significantly reduce computational overhead. This would greatly improve efficiency. We presents the details of our Thompson Sampling algorithm in Appendix 1.

## 5 EXPERIMENTS

Our experiments consist of three main parts: **Experiment 1**: Evaluate different methods on nine public reoptimization datasets, focusing on whether they can quickly find feasible solutions within the 10-second time limit. **Experiment 2**: Assess the quality of the feasible solutions obtained within the 10-second limit. **Experiment 3**: To provide a more intuitive comparison of solution convergence speeds, we plot the relative primal gap over time under a larger time limit of 100-second.

### 5.1 EXPERIMENTAL SETUP

**Benchmarks.** We select 9 series of instances from the MIP Computational Competition 2023 (Bolusani et al., 2023) to evaluate our approach. Each series has 50 similar instances with one or more components changing across instances. These instances need SCIP to solve from 60 to 600 seconds. Depending on the series, one of the following input can vary: (1) objective function coefficients (**obj_1, obj_2**), (2) variable bounds (**bnd_1, bnd_2, bnd_3**), (3) constraint right-hand sides (**rhs_1, rhs_2, rhs_3**), (4) constraint coefficients (**mat_1**). Most of these series are based on instances from the MIPLIB 2017 benchmark library (Gleixner et al., 2021) and some of others are collected from the real-world industrial use case and traditional problems. Due to limited space, please see Appendix C.1 for details of these datasets.

**Baselines.** We compared our approach against four baselines: the state-of-the-art open-source solver **SCIP** (Bestuzheva et al., 2021), the leading reoptimization method **Re_Tuning** (Patel, 2024), which won first place at the MIP Workshop 2023 competition (Bolusani et al., 2023) and does not rely on machine learning, and two GNN-based machine learning methods. Specifically, **PS** (Han et al., 2023) is primarily based on the large neighborhood search (LNS) method, while **ND** (Nair et al., 2020) utilizes a variable-fixing strategy for optimization. Please see Appendix C.2 for implementation details of these baselines. We also provide results for SCIP using the base solution as a warm-start strategy. This additional results is detailed in Appendix C.7.

**Training.** Each dataset contains 50 instances. To facilitate the experiments, we pair the instances in groups of two, resulting in 25 groups, including 20 groups in the training set and 5 groups in the test set. The first instance in each group serves as the base instance, for which intermediate solving information required for feature extraction is pre-recorded. The specific features are detailed in Table 5, in the Appendix A. All numerical results are reported for the test set. The model was implemented in PyTorch (Paszke et al., 2019) and optimized using Adam (Kingma & Ba, 2014) with training batch size of 16. The training process is conducted on a single machine that contains eight GPU devices (NVIDIA GeForce RTX 4090) and two AMD EPYC 7763 CPUs.

Table 3: Number of solved problems within 10s time limit for each method across datasets.

| Methods | bnd_1 | bnd_2 | bnd_3 | obj_1 | obj_2 | mat_1 | rhs_1 | rhs_2 | rhs_3 |
|---|---|---|---|---|---|---|---|---|---|
| SCIP | 5/5 | 0/5 | 0/5 | 5/5 | 5/5 | 5/5 | 5/5 | 5/5 | 5/5 |
| ND | 0/5 | 0/5 | 0/5 | 5/5 | 5/5 | 0/5 | 0/5 | 0/5 | 0/5 |
| PS | 5/5 | 0/5 | 0/5 | 5/5 | 5/5 | 5/5 | 5/5 | 5/5 | 5/5 |
| Re_Tuning | 5/5 | 3/5 | 3/5 | 5/5 | 5/5 | 5/5 | 5/5 | 5/5 | 5/5 |
| VP-OR(Ours) | 5/5 | 5/5 | 5/5 | 5/5 | 5/5 | 5/5 | 5/5 | 5/5 | 5/5 |

**Evaluation Metrics.** For each instance, we first solve the problem without a time limit and record the optimal solution's objective value as $OPT$. Then, we apply a time limit of 10 seconds for each method. The best objective value obtained within the time limit is denoted as $OBJ$. We define the following performance metrics: **(1) Solve Number**: This is the most fundamental metric, tracking the number of times a method successfully finds a feasible integer solution within the 10-second time limit. **(2) Gap**: We define the absolute and relative primal gaps as: **gap_abs** $= |OBJ - OPT|$ and **gap_rel** $= |OBJ - OPT|/(|OPT| + 10^{-10})$, respectively, and use them as performance metrics. Clearly, a smaller primal gap indicates a stronger performance. **(3) Wins**: This metric counts the number of instances where each method achieved the closest solution to the optimal one within the same time limit, relative to the total number of instances.

Throughout all experiments, we use SCIP 8.0.4 (Bestuzheva et al., 2021) as the backend solver, which is the state-of-the art open source solver, and is widely used in research of machine learning for combinatorial optimization (Chmiela et al., 2021; Khalil et al., 2022; Gasse et al., 2019). We keep all the other SCIP parameters to default and emphasize that all of the SCIP solver's advanced features, such as presolve and heuristics, are open.

## 5.2 EXPERIMENTAL RESULTS

In our experiments, we include only one parameter: the percentage of fixed variables $P$. In this section, we present the results for $P = 0.7$. Results for other values of $P$ are provided in Appendix C.3.

**Experiment 1.** The results in Table 3 show how each method performs under the 10-second time constraint to find an integer feasible solution, which reflects the real-world need for quickly obtaining high-quality reoptimization solutions (Marcucci & Tedrake, 2020; Zhang et al., 2020). We observe that only our method, VP-OR successfully found feasible solutions across all datasets within the time limit. The reoptimization method, Re_Tuning, also performed relatively well compared to other methods. This improved performance can be attributed to its use of warm-starting with solutions from previous instances and parameter tuning using historical solving information.

**Experiment 2.** We evaluate the quality of the best feasible solutions found by different methods within the 10-second time limit. The evaluation is conducted across various datasets, with performance measured by absolute gap (*gap_abs*), relative gap (*gap_rel*), and the number of wins (*wins*), where wins indicate the number of datasets for which a method achieves the best solution. The results are shown in Table 4, where "-" represents cases where the method could not find a feasible solution. In terms of *wins* and *gap_rel*, VP-OR surpasses all baseline methods. VP-OR performs exceptionally well in scenarios involving changes to variable bounds, matrix coefficients, and constraint right-hand sides. Specifically, in datasets where variable bounds are altered (e.g., *bnd_2*, and *bnd_3*), VP-OR achieves the average relative gap close to 0.1 in 10 seconds, while other methods struggle to provide feasible solutions within 100 seconds. Additionally, Re_Tuning outperforms both SCIP and end-to-end prediction-based methods on most datasets. ND and PS might be more suitable for problems that are not time-sensitive and allow for longer solving times.

**Experiment 3.** To provide a more intuitive comparison of solution convergence speeds, we plot the relative primal gap over time with a larger time limit of 100 seconds, highlighting how our approach converges compared to other methods. We observe that VP-OR is more suitable for scenarios that require rapidly obtaining high-quality solutions in the short term. It converges quickly to find high-quality feasible solutions in the early stages of solving, but in the global scope, we also found that our method may encounter the possibility of getting stuck at suboptimal solutions. While Re_Tuning and LNS also show potential, it's noteworthy that in certain cases, SCIP performs even better than some of the optimization methods. Due to space constraints, we only present the results from three datasets in this section, with additional results provided in Appendix C.4.

Table 4: Policy evaluation on the datasets, where "-" represents cases where the method could not find a feasible solution. The best performance is marked in bold.

| Methods | bnd_1 | | | bnd_2 | | | bnd_3 | | |
|---------|---------|---------|------|---------|---------|------|---------|---------|------|
| | gap_abs | gap_rel | wins | gap_abs | gap_rel | wins | gap_abs | gap_rel | wins |
| SCIP | 1974.20 | 0.16 | 1/5 | - | - | - | - | - | - |
| ND | - | - | - | - | - | - | - | - | - |
| PS | 9665.20 | 0.81 | 0/5 | - | - | - | - | - | - |
| Re_Tuning | 1425.5 | 0.12 | 0/5 | - | - | - | - | - | - |
| VP-OR(Ours) | **299.40** | **0.02** | **4/5** | **40.20** | **0.11** | **5/5** | **28.60** | **0.06** | **5/5** |

| Methods | mat_1 | | | obj_1 | | | obj_2 | | |
|---------|---------|---------|------|---------|---------|------|---------|---------|------|
| | gap_abs | gap_rel | wins | gap_abs | gap_rel | wins | gap_abs | gap_rel | wins |
| SCIP | 14.10 | 0.23 | 0/5 | 11.40 | 0.00 | 0/5 | 626.52 | 0.39 | 0/5 |
| ND | - | - | - | 11.40 | 0.00 | 0/5 | 674.21 | 0.44 | 0/5 |
| PS | 14.10 | 0.23 | 0/5 | 13.40 | 0.00 | 0/5 | 397.53 | 0.51 | 0/5 |
| Re_Tuning | 30.06 | 0.48 | 0/5 | 10.25 | 0.00 | 0/5 | **74.10** | 0.09 | 1/5 |
| VP-OR(Ours) | **10.09** | **0.16** | **5/5** | **3.28** | **0.00** | **5/5** | 329.99 | **0.06** | 4/5 |

| Methods | rhs_1 | | | rhs_2 | | | rhs_3 | | |
|---------|---------|---------|------|---------|---------|------|---------|---------|------|
| | gap_abs | gap_rel | wins | gap_abs | gap_rel | wins | gap_abs | gap_rel | wins |
| SCIP | 173.08 | 0.50 | 0/5 | 12.29 | 0.00 | 0/5 | 15.01 | 0.00 | 0/5 |
| ND | - | - | - | - | - | - | - | - | - |
| PS | 67090.50 | 193.04 | 0/5 | 22.25 | 0.00 | 0/5 | 18.00 | 0.00 | 0/5 |
| Re_Tuning | 6.40 | 0.02 | 0/5 | 2.24 | 0.00 | 0/5 | 0.40 | 0.00 | 0/5 |
| VP-OR(Ours) | **0.73** | **0.00** | **5/5** | **1.85** | **0.00** | **5/5** | **0.26** | **0.00** | **5/5** |

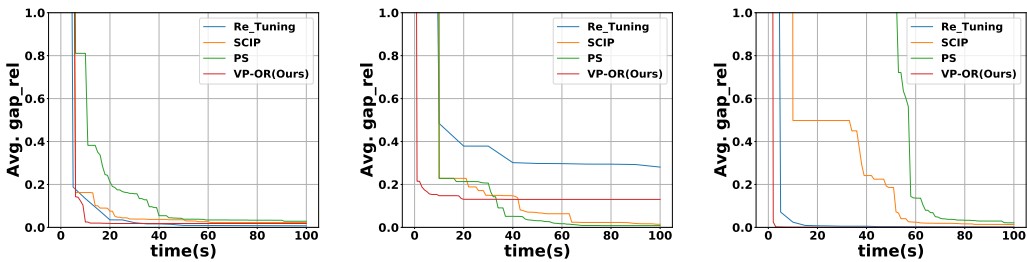

Figure 2: Performance comparisons in bnd_1, mat_1 and rhs_1, where the y-axis is average relative primal gap; each plot represents one benchmark dataset.

## 6 CONCLUSION

This paper proposes VP-OR, a two-stage reoptimization framework for MILPs with dynamic parameters. VP-OR first trains a GNN model to predict the marginal probability of each binary variable and the feasible ranges of integer and continuous variables in the modified MILP instance. Further, the Thompson Sampling algorithm is employed to iteratively select which variables to apply the predicted intervals, and adjust the marginal probability of each binary variable, ultimately solving for near-optimal solutions. Experimental evaluations conducted on 9 MILP datasets demonstrate that our framework outperforms four baselines.

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

## A  MORE DETAILS OF GRAPH FEATURES.

The feature extraction process is split into two parts: base instance and modified instance. For the base instance, we extract a richer set of graph features, including intermediate solving information. For the modified instance, we focus on structural information as Gasse et al. (2019). A list of the features used in our graph representation of the base instance is detailed in Table 5.

Table 5: Description of the constraint, variable and edge features in our graph representation of the base instance.

| Category | Feature | Description |
|---|---|---|
| variable vertex | lb | Original lower bound. |
| | ub | Original upper bound. |
| | objective_coeff | Objective coefficient. |
| | var_type | Type (binary, integer and continuous) as a one-hot encoding. |
| | leaf_lb | Lower bound of the leaf node which contains the optimal solution. |
| | leaf_ub | Upper bound of the leaf node which contains the optimal solution. |
| | depth | Depth of the leaf node. |
| | estimate | Estimate value of the leaf node. |
| | isBasic_var | If the variable is a basic variable in the LP relax of the leaf node. |
| | optvalue | Variable value in the optimal solution. |
| constraint edge | coef | Constraint coefficient. |
| constraint vertex | rhs | Right-hand side of the constraint. |
| | cons_type | Constraint type feature (eq, geq) as a one-hot encoding. |
| | isBasic_cons | If the constraint is a basic vector in the LP relax of the leaf node. |

## B  MORE DETAILS OF THOMPSON SAMPLING ALGORITHM

The Parameter Update Process algorithm is designed to update the parameters of the pior distributions for binary and continuous/integer variables based on the outcomes of the current solution. The goal is to refine these parameters to improve the performance of the Thompson Sampling approach in subsequent iterations. We adjust our fixing strategy using the Beta distribution parameters, $\alpha$ and $\beta$. The mean of the $Beta(\alpha, \beta)$ distribution is $\frac{\alpha}{\alpha+\beta}$. As these parameters increase, the distribution becomes more concentrated around the mean. With a prior of $Beta(\alpha, \beta)$, the posterior updates to $Beta(\alpha + 1, \beta)$ or $Beta(\alpha, \beta + 1)$.

In each iteration, we fix a percentage a% of the variables. When we find a better solution, we update the Beta distribution for the remaining 1-a% of unfixed variables based on this new solution. We also compare the current strategy to the one from the previous round that gave the best solution. If

a variable was fixed before but left unfixed in the current iteration, it indicates the previous strategy limited the solution quality. We update the Beta distributions for these variables to reflect this. In the next round, we resample the fixing strategy using these updated Beta distributions.

---

**Algorithm 2** Parameter Update Process

---

1: **Input:** Current solution $x_t$, Best solution $x^*$ from previous iterations, Best objective value $z^*$
2: **if** $z_t$ is better than $z^*$ **then**
3:      Set $r_t = 1$
4:      Update the best solution $x_t^* = x_t$
5:      Update priors for binary variables $i$:
6:      **for** each unselected binary variable $i$ **do**
7:          **if** $x_i = 1$ **then**
8:              $\alpha_i \leftarrow \alpha_i + 1$
9:          **else**
10:              $\beta_i \leftarrow \beta_i + 1$
11:          **end if**
12:      **end for**
13:      **for** each selected binary variable $i$ **do**
14:          **if** $x_i = 1$ and $x^* = 0$ **then**
15:              $\alpha_i \leftarrow \alpha_i + 1$
16:          **else if** $x_i = 0$ and $x^* = 1$ **then**
17:              $\beta_i \leftarrow \beta_i + 1$
18:          **end if**
19:      **end for**
20:      Update priors for continuous/integer variables $j$:
21:      **for** each unselected continuous/integer variable $j$ **do**
22:          **if** $x_j$ violates predicted bounds **then**
23:              $\beta_j \leftarrow \beta_j + 1$
24:          **else**
25:              $\alpha_j \leftarrow \alpha_j + 1$
26:          **end if**
27:      **end for**
28:      **for** each selected continuous/integer variable $j$ **do**
29:          **if** $x_j$ wasn't selected in the previous best solution $x_{t-1}^*$ **then**
30:              $\beta_j \leftarrow \beta_j + 1$
31:          **end if**
32:      **end for**
33: **else**
34:      Set $r_t = 0$
35: **end if**

---

When faced with infeasible instances, it typically indicates that some variable predictions are incorrect, resulting in conflicts with constraints. Our relaxation mechanism addresses this by dividing the conflicting variables into $G$ groups and subsequently solving each without these variable sets.

When a feasible solution cannot be found, we repeatedly apply the relaxation mechanism, building upon previous relaxations. Each iteration of this mechanism reduces the number of fixed variables. Therefore, theoretically, with enough iterations, we can ensure that the variables causing conflicts with the constraints are filtered out. However, in practice, it usually takes only a few iterations to obtain a feasible solution. For example, in the case of bnd_1, there are errors for only 8 for 2993 binary variables. By splitting these 8 erroneous variables into 10 groups, at least one group will inevitably exclude the erroneous variables. Of course, when the number of erroneous variables is greater, this is not guaranteed, but it is important to note that some variables, even if mispredicted, do not affect the ability to find a feasible solution due to their limited impact on solution sensitivity. We can easily filter out some variables that are highly sensitive to solution quality for each group.

---

**Algorithm 3** Relaxation Mechanism

---

1: **if** solution becomes infeasible **then**
2:     Divide fixed variables into $G$ groups.
3:     **for** each group $g = 1, 2, \ldots, G$ **do**
4:         Relax fixed variables in group $g$ back to their original bounds.
5:         Solve the subproblem with these relaxed constraints.
6:         **if** feasible solution found **then**
7:             Proceed to the next iteration.
8:         **end if**
9:     **end for**
10: **end if**

---

## C   MORE DETAILS OF EXPERIMENTS

### C.1   DATASETS

We selected the datasets based on two key considerations: first, the varying components within the instances, and second, the number of different variable types (integer, binary, continuous) present in each dataset. We aim for our evaluation to cover a wide range of variable types and varying components as comprehensively as possible.

The varying components of nine datasets are summarized in Table 6.

Table 6: The varying components of datasets.

| Datasets | Varying component | | | | | | Max Time |
|---|---|---|---|---|---|---|---|
| | LO | UP | OBJ | LHS | RHS | MAT | (s) |
| bnd_1 | ✓ | | | | | | 600 |
| bnd_2 | ✓ | ✓ | | | | | 300 |
| bnd_3 | ✓ | ✓ | | | | | 600 |
| obj_1 | | | ✓ | | | | 400 |
| obj_2 | | | ✓ | | | | 300 |
| mat_1 | | | | | | ✓ | 300 |
| rhs_1 | | | | ✓ | ✓ | | 400 |
| rhs_2 | | | | | ✓ | | 60 |
| rhs_3 | | | | | ✓ | | 60 |

We have shown the variable counts of some datasets in Table 1, and the remaining part is listed as follows. The details of each dataset is as follows:

Table 7: Comparison of variable prediction accuracy for the remaining part of datasets.

| Var. num. | bnd_2 | bnd_3 | rhs_3 |
|---|---|---|---|
| binary var. | 1457.0 | 1457.0 | 500.0 |
| mispredicted binary var. | 6.7 | 4.5 | 0.0 |
| integer var. | 0.0 | 0.0 | 0.0 |
| mispredicted integer var. | 0.0 | 0.0 | 0.0 |
| continuous var. | 301.0 | 301.0 | 500.0 |
| mispredicted continuous var. | 0.0 | 2.0 | 4.7 |

**bnd_1:** This dataset is from "bnd_s1" in the MIP Computational Competition 2023 (Bolusani et al., 2023). The instance is based on the instance *rococoC10-001000* from the MIPLIB 2017 benchmark library (Gleixner et al., 2021). The instances were generated by perturbing the upper bounds of general integer variables selected via a discrete uniform distribution up to ±100% of the bound value.

**bnd_2:** This dataset is from "bnd_s2" in the MIP Computational Competition 2023 (Bolusani et al., 2023). This series is based on the instance *csched007* from the MIPLIB 2017 benchmark library (Gleixner et al., 2021). The instances were generated via random fixings of 15% to 25% of the binary variables selected via a discrete uniform distribution w.r.t. the original instance.

**bnd_3:** This dataset is from "bnd_s3" in the MIP Computational Competition 2023 (Bolusani et al., 2023). This series is also based on the instance *csched007* from the MIPLIB 2017 benchmark library (Gleixner et al., 2021). The instances were generated via random fixings of 5% to 20% of the binary variables selected via a discrete uniform distribution w.r.t. the original instance. These instances are relatively harder to solve as compared to the instances in **bnd_2**.

**obj_1:** This dataset is from "obj_s1" in the MIP Computational Competition 2023 (Bolusani et al., 2023). This series is based on the stochastic multiple binary knapsack problem (Angulo et al., 2016). The problem is modeled as a two-stage stochastic MILP and one-third of the objective vector varying across instances.

**obj_2:** This dataset is from "obj_s2" in the MIP Computational Competition 2023 (Bolusani et al., 2023). The instances are based on the instance *ci-s4* from the MIPLIB 2017 benchmark library (Gleixner et al., 2021) with random perturbations and random rotations of the objective vector.

**mat_1:** This dataset is from "mat_s1" in the MIP Computational Competition 2023 (Bolusani et al., 2023). This series is based on the optimal vaccine allocation problem (Tanner & Ntaimo, 2010) and generated with varying constraint coefficients in the inequality constraints.

**rhs_1:** This dataset is from "rhs_s1" in the MIP Computational Competition 2023 (Bolusani et al., 2023). This series is based on the stochastic server location problem (Ntaimo, 2010). The instances is generated by the given dataset, and only the right-hand side vector of equality constraints varying across instances.

**rhs_2:** This dataset is from "rhs_s2" in the MIP Computational Competition 2023 (Bolusani et al., 2023). This series is based on a synthetic MILP and the associated dataset proposed by Jiménez-Cordero et al. (2022). The instances are generated by taking a convex combination of two different RHS vectors.

**rhs_3:** This dataset is from "rhs_s4" in the MIP Computational Competition 2023 (Bolusani et al., 2023). This series is also based on the synthetic MILP (Jiménez-Cordero et al., 2022). The instances are generated by taking a convex combination of two different RHS vectors(different than the ones used for generating **rhs_2**).

## C.2 Implementation Details of the Baselines

All baselines that provided open-source implementations, including PS and Re_Tuning, were tested using their official code. Since ND did not provide open-source code, we reproduced their method to the best of our ability based on their paper (Nair et al., 2020) and fine-tuned the parameters accordingly.

**SCIP.** We use SCIP 8.0.4 (Bestuzheva et al., 2021), which is the state-of-the art open source solver. We keep all the other SCIP parameters to default and emphasize that all of the SCIP solver's advanced features, such as presolve and heuristics, are open.

**Re_Tuning.** Re_Tuning is a state-of-the-art heuristic reoptimization framework (Patel, 2024), which does not utilize machine learning models. This framework, developed for the MIP 2023 workshop's computational competition (Bolusani et al., 2023), earned the first prize. It is primarily based on reusing historical branches and fine-tuning SCIP's parameters for more effective reoptimization. Our investigation revealed that Re_Tuning adjusts its configurations based on the previous instances it solves. Specifically, it may disable modules such as presolving or generating cutting planes for subsequent instances. While these adjustments have been shown to potentially improve overall solv-

ing time on certain datasets, they inevitably make it more challenging to find high-quality feasible solutions quickly in the early stages. To address this, we ensured these modules remained enabled for all instances, striving to achieve the best possible results with their code.

**Predict-and-Search(PS).** PS is an end-to-end machine learning-based approach (Han et al., 2023) which employs large neighborhood search (LNS) combined with GNN predictions. In practice, we do not know how many variables may be predicted incorrectly, and selecting an appropriate radius $\delta$ for the neighborhood in LNS can be time-consuming. To better demonstrate the performance of the PS method, we select the radius $\delta$ based on the average number of binary prediction errors observed during our preliminary tests, as shown in Table 1.

**Neural Diving(ND).** Another notable method we compared against is Neural Diving (**ND**) framework with Selective Net (Nair et al., 2020), which is also based on a variable-fixing strategy. Since ND focuses on fixing variables to accelerate the solving process, it serves as a relevant baseline to evaluate alongside our approach.

## C.3 MORE RESULTS WITH DIFFERENT PARAMETERS

In this section, we present a comprehensive evaluation of policy performance across various synthetic and real-world datasets, using different time and fix parameters. Each table below illustrates the impact of varying these parameters on the performance metrics, namely the absolute and relative gaps. The methods examined include SCIP, Re_Tuning, ND, and PS, alongside our proposed method, VP-OR, under different time constraints and fixed parameter ratios.

Table 8 illustrates the performance of various methods under different boundary conditions (bnd_1, bnd_2, bnd_3). After reoptimizing with adjusted boundary parameters, the VP-OR method consistently shows lower absolute and relative gaps compared to SCIP and other comparative methods under different time constraints (T=10 and T=20).

Table 8: Policy evaluation on the synthetic and real-world datasets with different time and fix parameters. We report the arithmetic mean of gap_abs and gap_rel.

| Methods | bnd_1 | | bnd_2 | | bnd_3 | |
|---|---|---|---|---|---|---|
| | gap_abs | gap_rel | gap_abs | gap_rel | gap_abs | gap_rel |
| SCIP (T=10.0) | 1974.20 | 0.16 | - | - | - | - |
| Re_Tuning (T=10.0) | 1425.5 | 0.12 | - | - | - | - |
| ND (T=10.0, P=0.5) | - | - | - | - | - | - |
| PS (T=10.0, P=0.5) | 9439.60 | 0.79 | - | - | - | - |
| VP-OR(Ours) (T=10.0, P=0.5) | 279.20 | 0.02 | 40.20 | 0.11 | 31.20 | 0.09 |
| ND (T=10.0, P=0.6) | - | - | - | - | - | - |
| PS (T=10.0, P=0.6) | 9439.60 | 0.79 | - | - | - | - |
| VP-OR(Ours) (T=10.0, P=0.6) | 528.80 | 0.04 | 39.40 | 0.11 | 37.40 | 0.11 |
| ND (T=10.0, P=0.7) | - | - | - | - | - | - |
| PS (T=10.0, P=0.7) | 9665.20 | 0.81 | - | - | - | - |
| VP-OR(Ours) (T=10.0, P=0.7) | 299.40 | 0.02 | 40.20 | 0.11 | 28.60 | 0.06 |
| ND (T=10.0, P=0.8) | - | - | - | - | - | - |
| PS (T=10.0, P=0.8) | 1216.80 | 0.10 | - | - | - | - |
| VP-OR(Ours) (T=10.0, P=0.8) | 973.80 | 0.08 | 38.80 | 0.11 | 34.80 | 0.10 |
| SCIP (T=20.0) | 921.00 | 0.08 | - | - | - | - |
| Re_Tuning (T=20.0) | 402.25 | 0.03 | 26.0 | 0.06 | - | - |
| ND (T=20.0, P=0.5) | - | - | - | - | - | - |
| PS (T=20.0, P=0.5) | 2483.00 | 0.20 | - | - | - | - |
| VP-OR(Ours) (T=20.0, P=0.5) | 313.20 | 0.03 | 48.60 | 0.14 | 33.80 | 0.10 |
| ND (T=20.0, P=0.6) | - | - | - | - | - | - |
| PS (T=20.0, P=0.6) | 2408.00 | 0.19 | - | - | - | - |
| VP-OR(Ours) (T=20.0, P=0.6) | 264.60 | 0.02 | 40.40 | 0.12 | 37.40 | 0.11 |
| ND (T=20.0, P=0.7) | - | - | - | - | - | - |
| PS (T=20.0, P=0.7) | 2627.40 | 0.21 | - | - | - | - |
| VP-OR(Ours) (T=20.0, P=0.7) | 299.40 | 0.02 | 39.80 | 0.11 | 23.40 | 0.07 |
| ND (T=20.0, P=0.8) | - | - | - | - | - | - |
| PS (T=20.0, P=0.8) | 1007.20 | 0.08 | - | - | - | - |
| VP-OR(Ours) (T=20.0, P=0.8) | 1409.40 | 0.12 | 41.40 | 0.12 | 23.40 | 0.07 |

Table 9 evaluates performance under different matrix and objective function settings (mat_1, obj_1, obj_2). With these adjustments, the VP-OR method maintains significant suppression of gap_abs and gap_rel, particularly excelling in objective function cases (obj_1 and obj_2).

Table 9: Policy evaluation on the synthetic and real-world datasets with different time and fix parameters. We report the arithmetic mean of gap_abs and gap_rel.

| Methods | mat_1 | | obj_1 | | obj_2 | |
|---|---|---|---|---|---|---|
| | gap_abs | gap_rel | gap_abs | gap_rel | gap_abs | gap_rel |
| SCIP (T=10.0) | 14.10 | 0.23 | 11.40 | 0.00 | 626.52 | 0.39 |
| Re_Tuning (T=10.0) | 30.06 | 0.48 | 10.25 | 0.00 | 74.10 | 0.09 |
| ND (T=10.0, P=0.5) | - | - | 11.40 | 0.00 | 634.70 | 0.39 |
| PS (T=10.0, P=0.5) | 14.10 | 0.23 | 13.40 | 0.00 | 387.89 | 0.51 |
| VP-OR(Ours) (T=10.0, P=0.5) | 9.09 | 0.15 | - | - | 7783.94 | 1.53 |
| ND (T=10.0, P=0.6) | - | - | 11.40 | 0.00 | 634.70 | 0.39 |
| PS (T=10.0, P=0.6) | 14.10 | 0.23 | 13.40 | 0.00 | 397.53 | 0.51 |
| VP-OR(Ours) (T=10.0, P=0.6) | 11.62 | 0.19 | - | - | 6854.66 | 0.75 |
| ND (T=10.0, P=0.7) | - | - | 11.40 | 0.00 | 674.21 | 0.44 |
| PS (T=10.0, P=0.7) | 14.10 | 0.23 | 13.40 | 0.00 | 397.53 | 0.51 |
| VP-OR(Ours) (T=10.0, P=0.7) | 10.09 | 0.16 | 3.28 | 0.00 | 329.99 | 0.06 |
| ND (T=10.0, P=0.8) | - | - | 11.40 | 0.00 | - | - |
| PS (T=10.0, P=0.8) | 14.10 | 0.23 | 13.40 | 0.00 | 702.68 | 0.41 |
| VP-OR(Ours) (T=10.0, P=0.8) | 11.77 | 0.19 | 338.60 | 0.04 | 8287.57 | 3.01 |
| SCIP (T=20.0) | 11.66 | 0.19 | 10.40 | 0.00 | 285.99 | 0.14 |
| Re_Tuning (T=20.0) | 18637.00 | 0.42 | 8.25 | 0.00 | 1.61 | 0.01 |
| ND (T=20.0, P=0.5) | - | - | 10.40 | 0.00 | 285.99 | 0.14 |
| PS (T=20.0, P=0.5) | 13.17 | 0.21 | 13.40 | 0.00 | 243.40 | 0.30 |
| VP-OR(Ours) (T=20.0, P=0.5) | 7.69 | 0.12 | - | - | 6855.85 | 0.75 |
| ND (T=20.0, P=0.6) | - | - | 10.40 | 0.00 | 268.18 | 0.13 |
| PS (T=20.0, P=0.6) | 13.17 | 0.21 | 13.40 | 0.00 | 243.40 | 0.30 |
| VP-OR(Ours) (T=20.0, P=0.6) | 9.94 | 0.16 | 19.40 | 0.00 | 6058.61 | 0.50 |
| ND (T=20.0, P=0.7) | - | - | 10.40 | 0.00 | 285.99 | 0.14 |
| PS (T=20.0, P=0.7) | 13.17 | 0.21 | 13.40 | 0.00 | 239.79 | 0.28 |
| VP-OR(Ours) (T=20.0, P=0.7) | 10.09 | 0.16 | 3.28 | 0.00 | 322.85 | 0.01 |
| ND (T=20.0, P=0.8) | - | - | 10.40 | 0.00 | - | - |
| PS (T=20.0, P=0.8) | 13.17 | 0.21 | 13.40 | 0.00 | 202.44 | 0.16 |
| VP-OR(Ours) (T=20.0, P=0.8) | 11.61 | 0.19 | 142.40 | 0.02 | 322.85 | 0.01 |

Table 10 shows the response of each method when adjusting the parameters on the right-hand side of constraints (rhs_1, rhs_2, rhs_3). In these scenarios, the VP-OR method achieves gaps close to zero.

Table 10: Policy evaluation on the synthetic and real-world datasets with different time and fix parameters. We report the arithmetic mean of gap_abs and gap_rel.

| Methods | rhs_1 | | rhs_2 | | rhs_3 | |
|---|---|---|---|---|---|---|
| | gap_abs | gap_rel | gap_abs | gap_rel | gap_abs | gap_rel |
| SCIP (T=10.0) | 173.08 | 0.50 | 12.29 | 0.00 | 16.77 | 0.00 |
| Re_Tuning (T=10.0) | 6.40 | 0.02 | 2.24 | 0.00 | 0.40 | 0.00 |
| ND (T=10.0, P=0.5) | - | - | - | - | - | - |
| PS (T=10.0, P=0.5) | 57558.07 | 165.41 | 13.23 | 0.00 | 12.46 | 0.00 |
| VP-OR(Ours) (T=10.0, P=0.5) | 0.27 | 0.00 | 1.85 | 0.00 | 0.26 | 0.00 |
| ND (T=10.0, P=0.6) | - | - | - | - | - | - |
| PS (T=10.0, P=0.6) | 62046.33 | 177.93 | 13.23 | 0.00 | 12.46 | 0.00 |
| VP-OR(Ours) (T=10.0, P=0.6) | 0.50 | 0.00 | 1.85 | 0.00 | 0.26 | 0.00 |
| ND (T=10.0, P=0.7) | - | - | - | - | - | - |
| PS (T=10.0, P=0.7) | 67090.50 | 193.04 | 22.25 | 0.00 | 17.98 | 0.00 |
| VP-OR(Ours) (T=10.0, P=0.7) | 0.73 | 0.00 | 1.85 | 0.00 | 0.26 | 0.00 |
| ND (T=10.0, P=0.8) | - | - | - | - | - | - |
| PS (T=10.0, P=0.8) | 66978.45 | 192.33 | 15.35 | 0.00 | 16.29 | 0.00 |
| VP-OR(Ours) (T=10.0, P=0.8) | 0.71 | 0.00 | 1.85 | 0.00 | 0.26 | 0.00 |
| SCIP (T=20.0) | 173.08 | 0.50 | 5.54 | 0.00 | 7.22 | 0.00 |
| Re_Tuning (T=20.0) | 2.85 | 0.01 | 0.00 | 0.00 | 0.00 | 0.00 |
| ND (T=20.0, P=0.5) | - | - | - | - | - | - |
| PS (T=20.0, P=0.5) | 38275.26 | 109.85 | 5.54 | 0.00 | 5.97 | 0.00 |
| VP-OR(Ours) (T=20.0, P=0.5) | 0.39 | 0.00 | 1.85 | 0.00 | 0.26 | 0.00 |
| ND (T=20.0, P=0.6) | - | - | - | - | - | - |
| PS (T=20.0, P=0.6) | 38275.26 | 109.85 | 5.47 | 0.00 | 5.97 | 0.00 |
| VP-OR(Ours) (T=20.0, P=0.6) | 0.26 | 0.00 | 1.85 | 0.00 | 0.26 | 0.00 |
| ND (T=20.0, P=0.7) | - | - | - | - | - | - |
| PS (T=20.0, P=0.7) | 65141.18 | 187.43 | 4.42 | 0.00 | 5.97 | 0.00 |
| VP-OR(Ours) (T=20.0, P=0.7) | 0.29 | 0.00 | 1.85 | 0.00 | 0.26 | 0.00 |
| ND (T=20.0, P=0.8) | - | - | - | - | - | - |
| PS (T=20.0, P=0.8) | 54004.10 | 155.03 | 3.30 | 0.00 | 2.16 | 0.00 |
| VP-OR(Ours) (T=20.0, P=0.8) | 0.40 | 0.00 | 1.85 | 0.00 | 0.26 | 0.00 |

## C.4 MORE RESULTS OF THE RELATIVE GAP

In the main text, we presented results for the datasets bnd_1, mat_1, and rhs_1. Here, we extend our analysis by providing additional results for the remaining datasets. This section focuses on performance comparisons in terms of the average relative gap *gap_rel*.

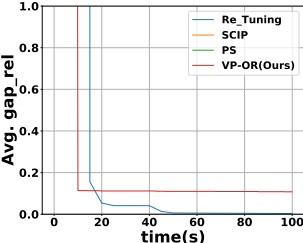 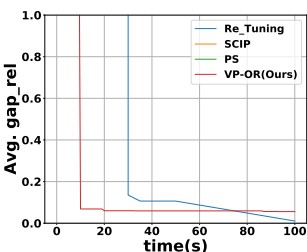 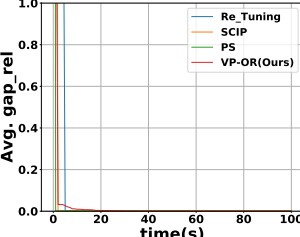

Figure 3: Performance comparisons in bnd_2, bnd_3 and obj_1, where the y-axis is average relative primal gap; each plot represents one benchmark dataset.

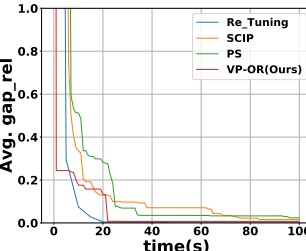 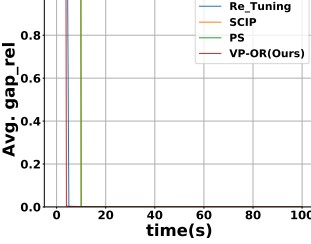 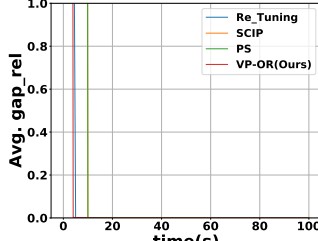

Figure 4: Performance comparisons in obj_2, rhs_2 and rhs_3, where the y-axis is average relative primal gap; each plot represents one benchmark dataset.

## C.5 ABLATION STUDY OF PREDICTION.

The table below demonstrates the predictive performance of both traditional Graph Neural Networks (GNN) and our approach in a reoptimization context(Re_GNN):

Table 11: Predictive performance of traditional Graph Neural Networks (GNN) and our prediction method in a reoptimization context(Re_GNN).

|  | bnd_1 | bnd_2 | bnd_3 |
|---|---|---|---|
| Total binary var. | 1457.0 | 1457.0 | 1457.0 |
| mispredicted binary var. (GNN) | 163.0 | 45.1 | 42.2 |
| mispredicted binary var. (Re_GNN) | **8.2** | **6.7** | **4.5** |
| Total integer var. | 124.0 | 0.0 | 0.0 |
| mispredicted integer var. (GNN) | 33.4 | 0.0 | 0.0 |
| mispredicted integer var. (Re_GNN) | **17.4** | 0.0 | 0.0 |
| Total continuous var. | 0.0 | 301.0 | 301.0 |
| mispredicted continuous var. (GNN) | 0.0 | 140.2 | 121.0 |
| mispredicted continuous var. (Re_GNN) | **0.0** | **0.0** | **2.0** |

## C.6 COMPUTATIONAL COMPLEXITY ANALYSIS

The primary computational complexity of VP-OR arises from the Thompson Sampling process. In the sampling phase of Thompson Sampling, we sample the probability $p$ for binary variables and select a certain percentage (a%) of variables based on the value of $min(p, 1 - p)$ by sorting them. For integer and continuous variables, we sample to determine whether they should be fixed and select the top a% of variables based on this criterion. This step has a time complexity of $O(nlogn)$, where n is the number of variables.

In the parameter update phase of Thompson Sampling, we update the parameters for each variable once. This step has a time complexity of $O(n)$. We tested the sampling time and parameter update time for each dataset, as presented in Table 12.

Table 12: Variable Numbers, Sampling Time(Time_s), and Parameter Update Time(Time_u) for Different Datasets

|  | bnd_1 | bnd_2 | bnd_3 | mat_1 | obj_1 | obj_2 | rhs_1 | rhs_2 | rhs_3 |
|---|---|---|---|---|---|---|---|---|---|
| **Var. num** | 3117 | 1758 | 1758 | 802 | 360 | 745 | 12760 | 1000 | 1000 |
| **Time_s (s)** | 0.008 | 0.005 | 0.003 | 0.013 | 0.002 | 0.016 | 0.102 | 0.002 | 0.002 |
| **Time_u (s)** | 0.002 | 0.001 | 0.000 | 0.009 | 0.001 | 0.010 | 0.070 | 0.001 | 0.001 |

## C.7 RESULTS FOR THE IMPACT OF INITIAL HINTS

We provided the initial solution as a hint for the "completesol" heuristic method during the presolve phase, effectively employing the base solution as a warm start(WS). We observe that SCIP has improvements in the quality of feasible solutions under these conditions. The results are shown in Table 13, where "-" indicates cases where the method could not find a feasible solution within the designated time limit.

Table 13: Performance Comparison Across SCIP, SCIP(WS) and VP-OR. We report the arithmetic mean of the metric gap_rel.

|  | bnd_1 | bnd_2 | bnd_3 | mat_1 | obj_1 | obj_2 | rhs_1 | rhs_2 | rhs_3 |
|---|---|---|---|---|---|---|---|---|---|
| **SCIP** | 0.16 | - | - | 0.23 | 0.00 | 0.39 | 0.50 | 0.00 | 0.00 |
| **SCIP(WS)** | 0.10 | - | - | 0.22 | 0.00 | 0.12 | 0.50 | 0.00 | 0.00 |
| **VP-OR(Ours)** | **0.02** | **0.11** | **0.06** | **0.16** | **0.00** | **0.06** | **0.00** | **0.00** | **0.00** |

## C.8 MORE RESULTS FOR EXPANDED TEST SAMPLES

The publicly available dataset from the MIP Workshop 2023 Computational Competition on Reoptimization (Bolusani et al., 2023) is limited in size, providing only 50 examples per task. To further increase the number of test samples, we attempt to generate similar datasets for testing by using a method consistent with the one published by the competition organizers. This step proves to be very time-consuming because random perturbations in the parameters often result in infeasible problems. During the dataset generation process, we repeatedly generate instances randomly until we find one that is feasible. Using the bnd_1 dataset as an example, we generate 100 additional instances. The results presented in the table below are consistent with our previous tests.

## C.9 MORE RESULTS FOR END-TO-END METHODS

Several end-to-end methods have been developed specifically for large-scale problems, such as GNN&GBDT (Ye et al., 2023) and Light-MILPopt (Ye et al., 2024). We conduct an experiment on the latest approach, Light-MILPopt. We observe that Light-MILPopt uses a variable fixing strategy, initially fixing k% of the variables based on predicted values (using the default setting k=20

Table 14: Policy evaluation on the bnd_1 dataset with 100 samples. We provide the metrics Average Relative Gap (gap_rel) and Average Absolute Gap (gap_abs).

|             | gap_rel | gap_abs |
|-------------|---------|---------|
| SCIP        | 0.20    | 2354.2  |
| PS          | 1.13    | 13213.0 |
| VP-OR(Ours) | **0.01** | **167.3** |

as per the authors' code). However, in a reoptimization context, fixing these variables often led to infeasibility in most instances. This is mainly because the model inaccurately predicts some variables, even when considered high-confidence. Consequently, we test the results with the variable fixing module disabled. The final experimental results present the number of instances that can find feasible solutions within a 10-second time limit in Table 15. Although this method is not specifically designed for reoptimization scenarios, which often demand rapid responses to slight changes in parameters with time-critical requirements for solutions, it does show some improvement over SCIP on more challenging datasets like bnd_2 and bnd_3.

We provide the average relative gap (gap_rel) for comparison in Table 16, where "-" represents cases where the method could not find a feasible solution within the time limit.

Table 15: Number of Instances Finding Feasible Solutions within 10 Seconds.

|                                      | bnd_1 | bnd_2 | bnd_3 |
|--------------------------------------|-------|-------|-------|
| SCIP                                 | 5/5   | 0/5   | 0/5   |
| Light-MILPopt                        | 0/5   | 0/5   | 0/5   |
| Light-MILPopt (without fix strategy) | 5/5   | 1/5   | 1/5   |
| VP-OR (Ours)                         | **5/5** | **5/5** | **5/5** |

Table 16: Average Relative Gap (gap_rel).

|                                      | bnd_1 | bnd_2 | bnd_3 |
|--------------------------------------|-------|-------|-------|
| SCIP                                 | 0.16  | -     | -     |
| Light-MILPopt                        | -     | -     | -     |
| Light-MILPopt (without fix strategy) | 0.22  | -     | -     |
| VP-OR (Ours)                         | **0.02** | **0.11** | **0.06** |

