# OpenReview forum: "A Reoptimization Framework for Mixed Integer Linear Programming with Dynamic Parameters"
_ICLR.cc/2025/Conference — Submitted to ICLR 2025_

### Official Review · Reviewer_6YTp · 2024-11-03

**Soundness:** 3
**Presentation:** 3
**Contribution:** 2
**Rating:** 5
**Confidence:** 4

**Summary:**

This paper provide a learning method for solving MILP instances that change slightly from previous solved instances (base instances). The paper leverage features from the solving process of the base instances to improve solving the modified instances. It uses a GNN to predict the solution values or the solution ranges. It also employs an iterative refinement process to refine the predictions by solving a multi-arm bandit problem. Evaluations are done on a sets of instances from the MIP 2023 computation competition. The runtime, solution quality are the main metrics for evaluation against other baselines.

**Strengths:**

1. The paper provides a novel ML method for MILP reoptimization. The innovation comes from leveraging the features from the solving process for the base instances and introducing the refinement methods for prediction confidence.
2. The paper handles MILP with not just binary variables, but also those with general integer and continuous variables. Engineering details are included in the methods to handle the those variables.
3. The paper is in general easy to follow.

**Weaknesses:**

1. My main criticism for the paper is that the instances used for evaluation are easy. It looks like the instances could be solved to close optimal in about 1 minute even with SCIP. The paper didn’t justify why easy instances are considered. In previous work, such as [1] (ND), [2] (PS) and [3] (a follow-up work of PS), much harder and larger instances are used in evaluation.
2. I am not sure if the main competitor Re_Tuning is properly implemented. It performs the worst in many case but it is the main competitor of the proposed methods in this paper.
3. While I understand the definition of the problem well, the paper doesn’t motivate the problem of reoptimization well. For example, why is this problem important? What are the main difference from the settings of [1,2,3]? Why do we consider instances that are easy that can be solved by SCIP to close-optimal within 60-80 seconds?

[1] Vinod Nair, et al. Solving mixed integer programs using neural networks. arXiv preprint, 2020

[2] Qingyu Han et al. A gnn-guided predict-and-search framework for mixed-integer linear programming. ICLR, 2023.

[3] Taoan Huang et al. Contrastive Predict-and-Search for Mixed Integer Linear Programs. ICML 2024.

**Questions:**

Please see the weaknessses.

---

> ### Author Response · Authors · 2024-11-25
> **Response to Reviewer 6YTp**
>
> ## Response to the weakness 1 and weakness 3
> Thank you for your insightful comments.
>
> **Dataset:** The datasets we tested are based on the MIP Workshop 2023 Computational Competition on Reoptimization [1]. These datasets primarily originate from MIPLIB [2] and some real-world time-sensitive reoptimization problems. MIPLIB is a well-known and widely used collection of benchmark problems in the field of mixed-integer linear programming (MILP). It is frequently maintained and updated to include a diverse set of test problems sourced from various real-world applications and industries.
>
> [1] Bolusani S, Besançon M, Gleixner A, et al. The mip workshop 2023 computational competition on reoptimization[J]. Mathematical Programming Computation, 2024: 1-12.
>
> [2] Gleixner A, Hendel G, Gamrath G, et al. MIPLIB 2017: data-driven compilation of the 6th mixed-integer programming library[J]. Mathematical Programming Computation, 2021, 13(3): 443-490.
>
> **About the reoptimization scenario:** We would like to emphasize the importance of reoptimization in many real-world scenarios, such as system control, railway scheduling, and production planning. These environments often require **rapid responses** to **slight changes** in parameters, with **time-critical demands** for solutions. In flexible industrial production settings, for example, each time a batch of products arrives at the warehouse, the solver must quickly determine routing and scheduling strategies in seconds. This involves coordinating different machines, including shuttles and stacker cranes, to move products onto shelves.  While the warehouse layout and list of machines remain constant, parameters related to shelf availability and machine operation status need adjustments to solve these tasks efficiently.
>
> **Problem setting:** While existing methods have explored solution predictions using end-to-end approaches, they typically do not **leverage historical information**, which limits their ability to meet time sensitivity requirements. As demonstrated in benchmarks like **bnd_2 and bnd_3**, other methods **struggle to provide feasible solutions within 100 seconds**, whereas our VP-OR approach can find solutions in under 10 seconds. The primary **difference** between our approach and existing methods is that we focus on reoptimization problems with dynamically changing parameters. Specifically, we consider scenarios similar to those described by Patel [3], involving a series of MILP instances. Each subsequent instance, known as the modified instance, is adapted from the previous instance, through random perturbations and adjustments to parameters such as the objective vector, constraints, and variable bounds. **The previous instances provide not only the optimal solution but also detailed records of intermediate computational steps, like selected branches and basis variables at each node’s LP relaxation.** These records can be strategically utilized in the reoptimization algorithm to accelerate the solving process for the modified instances.
>
> [3] Patel K K. Progressively strengthening and tuning MIP solvers for reoptimization[J]. Mathematical Programming Computation, 2024: 1-29.
> ## Response to the weakness 2
> We reproduced Re_Tuning by running the open-source code provided by the authors. However, we acknowledge your observation regarding the quality of feasible solutions found within a 10-second time limit. Our investigation revealed that Re_Tuning adjusts its configurations based on the previous instances it solves. Specifically, **it may disable modules such as presolving or generating cutting planes for subsequent instances**.
>
> While these adjustments have been shown to potentially improve overall solving time on certain datasets, they inevitably make it more challenging to find high-quality feasible solutions quickly in the early stages. To address this, we ensured these modules remained enabled for all instances, striving to achieve the best possible results with their code. The results are shown in the table below, where "NA" indicates cases where the method could not find a feasible solution within the designated time limit. We explain this in Appendix C.2 and replace the original version with the best results achieved by Re_Tuning in our experimental tables.
>
> | Method | bnd_1  | bnd_2 | bnd_3| mat_1|obj_1|obj_2|rhs_1|rhs_2|
> |-------|-------|-------|------|----|----|----|----|----|
> | SCIP |  0.16|NA |NA |0.23|0.00|0.39|0.50|0.00|0.00|
> | Re_Tuning |  0.13 |NA |NA |0.48 |0.01|0.09|0.02|0.00|0.01|
> | VP-OR(Ours) |  **0.02** |**0.11**|**0.06**  |**0.16**  |**0.00** |**0.06** |**0.00**|**0.00**|**0.00**|

---

> > ### Author Response · Authors · 2024-11-27
> >
> > Dear Reviewer 6YTp,
> >
> > With the rebuttal deadline approaching, we're eager to hear your thoughts on our response and revisions. Have we adequately addressed your concerns? Your feedback is crucial to us, and we're committed to further improving our paper based on your insights.
> >
> > Thank you for your time.
> >
> > Best regards,
> >
> > Authors

---

### Official Review · Reviewer_au9E · 2024-11-04

**Soundness:** 2
**Presentation:** 3
**Contribution:** 2
**Rating:** 5
**Confidence:** 5

**Summary:**

In this manuscript, the authors introduce a two-stage reoptimization framework designed to efficiently identify high-quality feasible solutions. This framework comprises an initial stage of variable prediction followed by an iterative online refinement process. The proposed methodology has been rigorously evaluated through extensive experimentation across nine diverse datasets, demonstrating its superiority over state-of-the-art methods and open-source solvers. These comprehensive evaluations highlight the framework's effectiveness and robustness across various scenarios.

**Strengths:**

The paper presents an interesting idea. Replacing LNS with the Thompson Sampling algorithm is an intriguing and valuable exploration. This approach effectively leverages real-time solution information and promotes greater exploration through sampling, enhancing the overall search process.

1.	Interesting Observations in Section 4.1: The observations presented in Section 4.1 are particularly intriguing. The detailed discussion of the mispredicted variables provides strong support for the design of the subsequent iterative online refinement process.

2.	Comprehensive Evaluation Datasets and convincing results: The selection of experimental datasets is commendably broad, incorporating a diverse array of modifications and updates to the initial problem. This breadth lends a significant degree of credibility to the reported outcomes. However, to further solidify the robustness and reliability of the findings, it is suggested that future work include experiments with more complex instances and comparisons against additional state-of-the-art (SOTA) baselines.

**Weaknesses:**

1.	The novelty appears somewhat limited. Moreover, the manuscript lacks a clear summary of its contributions and does not sufficiently differentiate itself from existing literature.

  e.g., in Section 3, it seems that the solution prediction framework closely resembles the approach proposed in ND (also proposed a method for handling general integer variables). The authors should provide a detailed comparison highlighting the distinctions between their method and ND.

2.	The experimental comparisons presented in the manuscript are insufficient. Notably, some of the latest works in the field have not been included for comparison, such as [1], [2], and [3].

  [1] Ye H, Xu H, Wang H, et al. GNN&GBDT-guided fast optimizing framework for large-scale integer programming[C]//International Conference on Machine Learning. PMLR, 2023: 39864-39878.

  [2] Ye H, Xu H, Wang H. Light-MILPopt: Solving Large-scale Mixed Integer Linear Programs with Lightweight Optimizer and Small-scale Training Dataset[C]//The Twelfth International Conference on Learning Representations.

  [3] Nair V, Alizadeh M. Neural large neighborhood search[C]//Learning Meets Combinatorial Algorithms at NeurIPS2020. 2020.

3.	ND and PS appear to perform significantly worse in terms of generating feasible solutions (Table 3). Is this due to suboptimal hyperparameter settings, such as the appropriate radius in PS? Could the authors clarify why their proposed method succeeds in finding feasible solutions in more instances?
4.	The rationale behind choosing Thompson Sampling in the "Iterative Online Refinement" section is not clearly articulated. What is the necessity of introducing Thompson Sampling?

  a)	Could the authors include additional ablation studies comparing the application of Thompson Sampling against traditional LNS strategies (e.g., those presented in [1, 2, 3] and PS)? This would help illustrate the contributions in this phase.

  b)	Important algorithm descriptions, such as Algorithm 1, should be moved to the main body of the paper since they are crucial for understanding the implementation of Thompson Sampling.

**Questions:**

1.	In Section 3.1, the rationale for distinguishing between base instances and modified instances should be clarified. Is this a common approach in reoptimization problems? Additionally, it would be beneficial to discuss the impact of this setting on the final results.
2.	Can the relaxation mechanism ensure the identification of feasible solutions? It would be beneficial to include a discussion or proof regarding the feasibility guarantees provided by the relaxation approach.
3.	On line 79, the author states that "LNS does not actually decrease the problem’s variable size." This assertion might not be entirely accurate, as specific variants of LNS can indeed significantly reduce the variable size at each iteration by fixing some variables, as seen in reference [4]. A similar concern applies to Table 2. The author should clearly indicate which particular LNS method is being discussed and explain the distinction between this method and the process of variable fixation.

  [4] Wu, Yaoxin, et al. "Learning large neighborhood search policy for integer programming." Advances in Neural Information Processing Systems 34 (2021): 30075-30087.

4.	It appears that only five test cases were utilized for each dataset. Given this limited sample size, it raises concerns about whether the results can adequately substantiate the effectiveness of the proposed method.

---

> ### Author Response · Authors · 2024-11-25
> **Response to Reviewer au9E (Part 1/3)**
>
> ## Response to the weakness 1
> Our method is indeed inspired by the ND framework [1], but there are significant differences between our approach and the ND method.
>
> **About integer variables:** You mentioned that the ND method can handle general integer variables, which is not entirely accurate. The ND method is primarily designed based on binary variables. Although they propose a conceptual extension to predict integer and continuous variables by **representing integer variables in binary form** and **predicting each bit's value**, they did not empirically test problems with integer and continuous variables. In our trials with their approach, we found that predicting values for most integer and continuous variables was inaccurate and prone to errors.  We conducted tests on three datasets, bnd_1, bnd_2, and bnd_3, and found that all predicted values differed from the actual optimal values. This discrepancy arises because inaccuracies in predicting certain bits of these variables led to overall errors in the predictions.
> |  | bnd_1  | bnd_2 | bnd_3 |
> |--|--|--|--|
> | Total integer var. |  124.0 | 0.0  | 0.0  |
> | mispredicted integer var. |  124.0 |0.0  | 0.0 |
> | Total continuous var. |  0.0 | 301.0 | 301.0 |
> | mispredicted continuous var. |  0.0 | 301.0 | 301.0 |
>
>
> **Variable bounds prediction:** However, **this does not mean the predictive results lack value**, so we proposed a method to predicting **the bounds of these variables**. Despite this adjustment, **without leveraging historical solution data**, our experiments indicated that even predicting bounds led to **a significant portion of variables being excluded from their optimal solutions**. The table below demonstrates the predictive performance of both traditional Graph Neural Networks (GNN) and our approach in a reoptimization context (Re_GNN):
>
> |  | bnd_1  | bnd_2 | bnd_3 |
> |--|--|--|--|
> | Total binary var. |  1457.0 | 1457.0  | 1457.0  |
> |mispredicted binary var. (GNN) |  163.0 | 45.1 | 42.2 |
> |mispredicted binary var. (Re_GNN)|  **8.2** |**6.7**  | **4.5**  |
> | Total integer var. |  124.0 | 0.0  | 0.0  |
> |mispredicted integer var. (GNN) |  33.4 |0.0  | 0.0 |
> |mispredicted integer var. (Re_GNN)|  **17.4** |0.0  | 0.0 |
> | Total continuous var. |  0.0 | 301.0 | 301.0 |
> |mispredicted continuous var. (GNN) |  0.0 |  140.2  |  121.0  |
> |mispredicted continuous var. (Re_GNN)|  **0.0** |**0.0**  | **2.0** |
>
> In our framework, we leverage historical solution instances and intermediate solving process information (e.g., the branches added at the final leaf node where the optimal solution is found, basis variables at each node’s LP relaxation) to aid our predictions. Results show that by incorporating historical solution process information, Re_GNN achieves more accurate variable predictions compared to traditional GNN methods.
>
> [1] Nair V, Bartunov S, Gimeno F, et al. Solving mixed integer programs using neural networks[J]. arXiv preprint arXiv:2012.13349, 2020.

---

> ### Author Response · Authors · 2024-11-25
> **Response to Reviewer au9E (Part 2/3)**
>
> ## Response to the weakness 2
> We sincerely appreciate your suggestion to consider the mentioned papers. First, we must clarify that these works address different problems than our study's focus. We reproduced the latest work among the mentioned papers, Light-MILPopt[2], using the authors’ open-source code. Subsequently, we will provide a detailed explanation of how our reoptimization scenario differs from traditional end-to-end approaches and present the results of Light-MILPopt.
>
> **About the reoptimization scenario:** Our VP-OR framework is specially designed for reoptimization scenarios to better suit real-world applications, such as system control, railway scheduling, and production planning. These environments often require **rapid responses** to **slight changes** in parameters, with **time-critical demands** for solutions. Our framework targets scenarios like flexible industrial production settings, where each batch arrival necessitates quick determination of routing and scheduling strategies within seconds. This involves dynamic coordination of machines while adapting to changes in shelf availability and machine status.
>
> **Results of Light_MILPopt:** We found that Light-MILPopt also employs a variable fixing strategy, where they initially fix k% of the variables based on predicted values (using the default setting k=20 as per the authors' code). However, in a reoptimization context, fixing these variables led to infeasibility in most of the instances. This occurs mainly because the model inaccurately predicts some variables, even though it considers them to be high-confidence. Consequently, we also tested the results with the variable fixing module disabled. The final experimental results present the number of instances that can find feasible solutions within a 10-second time limit:
>
> |  |bnd_1| bnd_2| bnd_3|
> |--|--|--|---|
> |SCIP|5/5|0/5 | 0/5|
> |PS|5/5|0/5 | 0/5|
> |Light-MILPopt|0/5|0/5| 0/5|
> |Light-MILPopt(without fix strategy)|5/5|1/5 |1/5|
> |VP-OR(Ours)|5/5|5/5 | 5/5|
>
> We also provided the average gap relative (gap_rel) for comparison, where "NA" indicates cases where the method could not find a feasible solution within the designated time limit(T=10s).
>
> |  | bnd_1| bnd_2| bnd_3|
> |--|--|--|--|
> |SCIP|0.16 |NA | NA|
> |PS|0.81|NA|NA|
> |Light-MILPopt|NA|NA|NA|
> |Light-MILPopt(without fix strategy)|0.22|NA|NA|
> |VP-OR(Ours)|**0.02**|**0.11**|**0.06**|
>
> [2] Ye H, Xu H, Wang H. Light-MILPopt: Solving Large-scale Mixed Integer Linear Programs with Lightweight Optimizer and Small-scale Training Dataset[C]//The Twelfth International Conference on Learning Representations.
>
> ## Response to the weakness 3
> **About the ND method:** ND relies on predictions of variables, which are used to fix values. However, if these predictions are inaccurate, this approach can lead to suboptimal or even infeasible solutions. In the ND method, the strategy involves pre-training a model to select which variables should be fixed and gradually reducing the number of fixed variables to ensure feasibility. Due to the ND method's lack of utilization of historical solution information, **the error rate in variable predictions is comparatively higher*. We found that even when gradually reducing the fixed variable size to 50%, a significant portion of datasets still fail to find feasible solutions. While the ND method shows significant advantages in overall solving time when based on SCIP, it does not necessarily outperform SCIP in finding higher-quality feasible solutions **under very strict time constraints**. This is because the time spent **exploring fixed variable sizes** can limit the quality of feasible solutions discovered within the remaining time.
>
> **About the PS method:** PS employs large neighborhood search (LNS) combined with GNN predictions. In practice, we do not know how many variables may be predicted incorrectly, and selecting an appropriate radius $\delta$ for the neighborhood in LNS can be time-consuming. To better demonstrate the performance of the PS method, **we select the radius $\delta$ based on the average number of binary prediction errors observed during our preliminary tests**. In our scenario, the PS method may not perform optimally because it is not designed for rapidly finding high-quality feasible solutions. Although LNS methods can reduce the constraint space and potentially increase the likelihood of finding better solutions in the early stages through presolving, they may also make the problem **more challenging to solve initially by imposing additional neighborhood constraints**.

---

> ### Author Response · Authors · 2024-11-25
> **Response to Reviewer au9E (Part 3/3)**
>
> ## Response to the weakness 4
> a) The experiment comparing Thompson Sampling against pure LNS strategies is impractical. As shown in Table 2, while LNS-based methods improve the overall solving time of SCIP, they do not reduce the variable size, and their impact on problem-solving is effective but limited. For instance, in the bnd_1 dataset, even with LNS methods, solving a single iteration takes **over 300 seconds**. Under a 10-second time constraint, pure LNS methods cannot complete the first iteration. Our Thompson Sampling approach is designed for **multi-round** iterative solving scenarios, where each round involves selecting a strategy for fixing variables. Therefore, Thompson Sampling cannot be applied in the context of pure LNS methods alone.
>
> b) Thank you for your suggestion. We have moved the key parts of the algorithm into the main body of the paper.
>
> ## Response to the question 1
> **Problem setting:** The setting is the same as the mip workshop 2023 computational competition on reoptimization[3] and Patel[4]. In the reoptimization scenario, involving a series of MILP instances based on an MILP (base instance) taken from a specific application. Each subsequent instance is modified from the previous one with random perturbations and rotations to parameters such as the objective vector, constraints, and variable bounds. The previous instances provide not only the optimal solution but also detailed records of intermediate computational steps, such as selected branches and basis variables at each node’s LP relaxation. These records can be strategically leveraged in the reoptimization algorithm to accelerate the solving process for the modified instances.
>
> **Impact of this setting:** The primary advantage lies in the variable prediction phase. Due to the ability to reuse intermediate information from previous intermediate computational steps, we can significantly enhance prediction accuracy (see the response to weakness 1).
>
> [3] Bolusani S, Besançon M, Gleixner A, et al. The mip workshop 2023 computational competition on reoptimization[J]. Mathematical Programming Computation, 2024: 1-12.
>
> [4] Patel K K. Progressively strengthening and tuning MIP solvers for reoptimization[J]. Mathematical Programming Computation, 2024: 1-29.
>
> ## Response to the question 2
> When faced with infeasible instances, it typically indicates that some variable predictions are incorrect, resulting in conflicts with constraints. Our relaxation mechanism addresses this by dividing the conflicting variables into ten groups and subsequently solving each without these variable sets.
>
> **Theoretical analysis**: When a feasible solution cannot be found, we repeatedly apply the relaxation mechanism, building upon previous relaxations. Each iteration of this mechanism **reduces the number of fixed variables**. Therefore, theoretically, with enough iterations, we can ensure that the variables causing conflicts with the constraints are filtered out.
>
> **In practice:** However, in practice, it usually takes only a few iterations to obtain a feasible solution. Firstly, it is important to note that some variables, even if mispredicted, do not affect the ability to find a feasible solution due to their limited impact on solution sensitivity. Secondly, even if it is necessary to filter out all inaccurate variables, the number of inaccurately predicted variables is not high due to the utilization of intermediate information from historical solution processes (see the response to the weakness 1). For example, in the case of bnd_1, there are inaccuracies for only 8 out of 2993 binary variables. By splitting these 8 inaccurate variables into 10 groups, at least one group will inevitably exclude the inaccurate variables.
>
> ## Response to the question 3
> In our paper, the LNS referred to is specifically pure LNS without the fix strategy.
> It is true that certain variants of LNS can incorporate fix strategies to reduce the variable size at each iteration. This actually supports our motivation; the fix strategy offers more advantages in reducing variable size compared to the pure LNS strategy.  We updated the description in the paper.
>
> ## Response to the question 4
> We utilized the publicly available dataset from the MIP Workshop 2023 Computational Competition on Reoptimization for testing. To further increase the number of test samples, we generated 100 additional instances on the bnd_s1 dataset using a method consistent with the one published by the competition organizers. The results in the table below are consistent with our previous tests.
>
> |    |  gap_rel |gap_abs|
> |-------|-------|---|
> | SCIP | 0.20  |2354.2|
> | PS |  1.13 | 13213.0|
> | VP-OR|  **0.01**  | **167.3**|

---

> > ### Author Response · Authors · 2024-11-27
> >
> > Dear Reviewer au9E,
> >
> > With the rebuttal deadline approaching, we're eager to hear your thoughts on our response and revisions. Have we adequately addressed your concerns? Your feedback is crucial to us, and we're committed to further improving our paper based on your insights.
> >
> > Thank you for your time.
> >
> > Best regards,
> >
> > Authors

---

> > > ### Comment · Reviewer_au9E · 2024-11-30
> > >
> > > Thank you for your detailed and thoughtful feedback. While I appreciate the thoroughness of the authors' response, I still have several concerns that I believe have not been fully addressed:
> > >
> > > 1. Differences from ND: The authors claim to propose a method for predicting variable bounds. However, ND also updates variable bounds using prediction results. The bound tightening based on predictions in ND appears quite similar to the methods described in this paper. Based on your descriptions, it seems that the primary difference in the "INITIAL VARIABLE PREDICTION" section is the set of features used. The authors mention that incorporating "historical solution data" improves prediction accuracy, but I view this as an incremental tuning specific to the task, as such historical data is often not available in general solving scenarios.
> > >
> > > 2. Comparison with Light-MILPopt: The authors emphasize that Light-MILPopt and other works address different problems than their study. However, even the ND and PS, which the authors compare against, are not specifically designed for reoptimization. I do not see this as a valid reason for excluding them from the comparison. Additionally, Light-MILPopt introduces a repair mechanism to handle infeasible predictions. Have the authors considered this repair mechanism? From the current results, it appears that Light-MILPopt is unable to find feasible solutions for any of the instances. Furthermore, could you please clarify what is meant by "without fix strategy"?
> > >
> > > 3. Relaxation Mechanism: Based on your description of the Relaxation Mechanism, it appears to address the issue of prediction inaccuracy, which may help in better finding feasible solutions. I am curious whether the tests for PS and ND included a similar relaxation mechanism. If not, would the comparison still be considered fair?
> > >
> > > 4. Simplicity of Instances: I concur with other reviewers that the tested instances are relatively simple, making it difficult to demonstrate the effectiveness of the proposed method in real-world reoptimization scenarios.

---

> ### Author Response · Authors · 2024-11-30
> **Follow-up Response to Reviewer au9E (Part 1/2)**
>
> Thank you for taking the time to read our responses. We address your newly raised concerns in order.
>
>
> ## **1. Differences from ND:**
> We must emphasize that the ND method [1] predicts the **actual values** of integer variables, rather than their bounds, which makes it fundamentally different from our approach. We will explain this in detail below. Firstly, to handle the situation where directly representing integer variables as binary results in too many bits, the ND method [1] introduces a hyperparameter $n_b$ that controls the maximum number of bits predicted for each variable along this bit sequence. They consider the first $n_b$ bits of an integer as the most significant bits during the solving process, which aligns with our idea. This supports our approach of predicting only the scale for each integer variable rather than its actual value. Your mention of their method handling bounds seems to originate from this statement in the ND paper [1]: "We train our model to predict the $n_b$ most significant bits of the value for $z$, given the upper and lower bounds for $z$." However, this statement refers only to predicting the initial $n_b$ bits of **the integer's value**, **without predicting the subsequent bits**, thereby creating bounds. They still do not address the issue of inaccuracies in predicting the first $n_b$ bits, which can lead to infeasible solutions.
>
> We would like to clarify that our focus is specifically on **reoptimization scenarios**, rather than general solving scenarios. In reoptimization, we need to provide high-quality solutions within strict time constraints, which requires us to better utilize historical information during the prediction phase to achieve more accurate prediction results.
>
>
> ## **2. Comparison with Light-MILPopt:**
>
> It is correct that even the ND [1] and PS [2], which we compare against, are not specifically designed for reoptimization. We found that these end-to-end methods, when applied directly to reoptimization scenarios, do not fully address the need for **rapid responses** to slight parameter changes, especially when time is critical for obtaining solutions. We compare ND [1] and PS [2] because ND [1] is an end-to-end method based on **fix strategies**, while PS [2] is primarily rooted **in an LNS framework**. These methods are **highly representative** in the realm of end-to-end approaches. Research in end-to-end learning-based methods is a very active field. Even noted approaches like Light-MILPopt [3] and GNN&GBDT [4] cannot replicate all end-to-end papers for comparison (for example, PS [2] is not compared), thus we do not see this as a valid critique.
>
> Additionally, following your suggestion, we replicated the Light-MILPopt work[3] **using the authors' original source code**. In the results of pure Light-MILPopt that we provided, we **made no modifications**, including the repair mechanism you mentioned.
> We found that Light-MILPopt's performance on reoptimization datasets is quite similar to PS. After investigating the repair mechanism carefully, we attempted to analyze why it did not make the problem feasible.
>
> The repair mechanism evaluates constraints that contain fixed variables by performing the following assessment: It calculates **the minimum possible value** of the left-hand side (LHS) of the constraint by substituting the bounds of all non-fixed variables into the expression. This calculated minimum is then compared to the right-hand side value $b_i$. If it determines that **the constraint cannot be satisfied under any bounds of the unfixed variables**, the fixed variables within this constraint are unfixed. Although this approach can effectively identify variables causing **individual constraint** infeasibility, in practical problems, many variables may create infeasibility due to **multiple combined constraints**. The repair mechanism cannot filter out these variables.
>
> **"Without fix strategy"** refers to the **variable reduction** module mentioned in Light-MILPopt (Section 3.3 of their paper [3]), where they initially fix k% of the variables based on predicted values, using the default setting of k=20 as per the authors' code. When there are prediction errors with high-confidence variables, it can easily lead to infeasibility.

---

> ### Author Response · Authors · 2024-11-30
> **Follow-up Response to Reviewer au9E (Part 2/2)**
>
> ## **3. Relaxation Mechanism:**
> The ND [1] and PS [2] methods also address situations where feasible solutions may not be found. Below, we provide a description of these methods and compare them with our relaxation mechanism.
>
> **The PS method** [2], based on pure LNS, can avoid infeasibility by appropriately selecting the neighborhood size $\delta$. This aligns with **the initial purpose of using the LNS framework** in the PS method, as mentioned in their paper [2]: "When fixing strategies may lead to suboptimal solutions or even infeasible sub-problems as a consequence of inappropriate ﬁxing, applying such a trust region search approach can always add ﬂexibility to the sub-problem".
>
> **The ND method** [1] gradually **reduces the scale of fixed variables** to ensure feasibility. Essentially, both the ND method [1] and our relaxation mechanism aim to avoid infeasibility by minimizing the set of fixed variables. However, there are notable **differences**: In their method, high-confidence variables are fixed following an initial prediction sort. In contrast, we treat variables with different prediction confidence levels equally, subdividing the set of fixed variables into multiple subsets to help eliminate inaccurately predicted variables.
>
> We do not agree that our comparison is unfair, as our relaxation mechanism is based on two key insights: our observations (p7, l327) that **high-probability variables can still be misclassified**, and findings in Table 1 that demonstrate integrating **historical information** significantly reduces prediction errors. However, we conducted an **ablation study** by applying our relaxation mechanism to the ND method to provide a better comparison.
>
> **Experimental results:** We tested ND with our relaxation mechanism (ND+relax), where "NA" indicates instances where the method could not find a feasible solution within the time limit (T=10s). We found that while most datasets still failed to yield feasible solutions, there was an improvement on certain datasets, such as obj_2, compared to the pure ND method. This is because **the original challenge remains across most datasets**: the time spent exploring fixed variable sizes can limit the quality of feasible solutions discovered within the remaining time.
>
>
> |  | bnd_1  | bnd_2 | bnd_3| mat_1|obj_1|obj_2|rhs_1|rhs_2|rhs_3|
> |-------|-------|-------|------|----|----|----|----|----|-----|
> | ND|NA|NA|NA|NA|0.00|0.44|NA|NA|NA|NA|
> | ND+relax |NA|NA|NA|NA|0.00|0.19 |NA|NA|NA|NA|
> | VP-OR(Ours) |**0.02**|**0.11**|**0.06**|**0.16**|**0.00**|**0.06**|**0.00**|**0.00**|**0.00**|
>
>
> ## **4. Datasets:**
> We do not agree that the scale of the instances we tested is unsuitable for **real-world reoptimization scenarios**, as we have mentioned in our response to Reviewer 6YTp. The datasets we tested are based on the **MIP Workshop 2023 Computational Competition on Reoptimization** [5].
> To the best our knowledge, this is the only dedicated reference benchmark for the reoptimization of MILPs.
> These datasets primarily originate from **MIPLIB** [6] and **some real-world time-sensitive reoptimization problems**. MIPLIB is a well-known and widely used collection of benchmark problems in the field of mixed-integer linear programming (MILP). It is frequently maintained and updated to include a diverse set of test problems sourced from **various real-world applications and industries**.
>
>
> Furthermore, we want to emphasize once again that Reviewer 6YTp considered our instances simple because they believed "the instances could be solved to near-optimality in about 1 minute even with SCIP". However, this is **not accurate**, as we have shown that on benchmarks like bnd_2 and bnd_3, other methods struggle to provide **even a feasible solutions within 100 seconds**, whereas our VP-OR approach can find solutions in under 10 seconds.
>
> ## **Reference:**
>
> [1] Nair V, Bartunov S, Gimeno F, et al. Solving mixed integer programs using neural networks[J]. arXiv preprint arXiv:2012.13349, 2020.
>
> [2] Han Q, Yang L, Chen Q, et al. A GNN-Guided Predict-and-Search Framework for Mixed-Integer Linear Programming[C]//The Eleventh International Conference on Learning Representations.
>
>
> [3] Ye H, Xu H, Wang H. Light-MILPopt: Solving Large-scale Mixed Integer Linear Programs with Lightweight Optimizer and Small-scale Training Dataset[C]//The Twelfth International Conference on Learning Representations.
>
>
> [4] Ye H, Xu H, Wang H, et al. GNN&GBDT-guided fast optimizing framework for large-scale integer programming[C]//International Conference on Machine Learning. PMLR, 2023: 39864-39878.
>
> [5] Bolusani S, Besançon M, Gleixner A, et al. The MIP Workshop 2023 Computational Competition on Reoptimization[J]. Mathematical Programming Computation, 2024: 1-12.
>
>
> [6] Gleixner A, Hendel G, Gamrath G, et al. MIPLIB 2017: Data-driven compilation of the 6th mixed-integer programming library[J]. Mathematical Programming Computation, 2021, 13(3): 443-490.

---

### Official Review · Reviewer_u68k · 2024-11-04

**Soundness:** 3
**Presentation:** 3
**Contribution:** 3
**Rating:** 8
**Confidence:** 4

**Summary:**

This paper proposes a two-stage learning-based framework for reoptimization of mixed-integer linear programs (MILPs), that is for solving MILP instances which do not differ in structure, but only with respect to parameter values (cost  coefficients, variable bounds, constraint coefficients and constraint RHS values). In the first stage, using instance features as well as features obtained from solving a base instance to optimality, GNNs are used to predict variable values (for binary variables) or ranges for variable values (for integer and continuous variables). In the second stage, these predictions are used to fix a sampled subset of the variables, and using Thompson sampling, the sampling is improved to obtain better feasible solutions.

In a set of computational experiments with a public reoptimization benchmark data set, the approach is compared against other reoptimization approaches as well as against other state-of-the art learning-based and learning-augmented approaches for solving MILPs. For a strict time limit of 10 seconds, the new approach mostly outperforms the other approaches, and for longer computation times, it shows a good convergence, in particular compared to other learning-based approaches.

**Strengths:**

1. Probably the closest approach to the presented paper is the predict-and-search (PS) approach by Han et al. (2023), which also can be viewed as a two-stage approach. Compared to that approach, this paper introduces two advancements: First, while the PS approach only predicts binary variable values, this approach also predicts ranges for general integer variables and continuous variables. Second, while PS uses a neighborhood search in the second stage, this approach cleverly combines variable fixing with Thompson sampling.

2. I find that both ideas (predicting ranges and using Thompson sampling for selecting variables (and bounds) to fix) form original contributions that nicely complement each other in this work, but that will be also useful for future works going beyond this paper.

3. The computational results in general are fairly strong; they show that in the reoptimization setting under very strict time limits (10 seconds) the approach outperforms the baseline approaches.

3. As far as I can see, the baselines are mostly reasonably chosen (with one exception).

**Weaknesses:**

1. When used in a reoptimization setting, it seems natural to warm-start SCIP with the base solution; this is apparently not performed in the paper. Neglecting this natural approach lets me think that maybe the results are a bit too favorable for the proposed approach, in particular given that SCIP is often the strongest contender in the experiments.

2.  On page 9, the authors write  "We observe that VP-OR consistently outperforms other methods, achieving the fastest convergence across datasets and rapidly closing the primal gap." Looking at Fig. 2, this statement is a bit optimistic, since e.g. in the middle plot, VP-OR never closes the gap, and other approaches close the gap at some point.




3. From the paper (Han et al 2023), it becomes clear that the PS approach only deals with binary variables. It does not become clear in the paper how that approach is adapted in the experiment to deal with instances also involing general integer variables and continuous variables.

4. The authors often use the term "x%", e.g. when referring to the fraction of variables to be fixed. As x is also used for decision variables, a different symbol would be better.

5. Fig. 2 is much too small, in particular w.r.t. the font sizes of the legends and axis labels.

6. The text mentions appendix D3 which is empty.

**Questions:**

1. I find that using plain SCIP in the reoptimization setting is a bit unfair. In general, in a reoptimization settign it would be reasonable to provide the base solution as a warm-start solution. I suggest you to include that as an additional baseline as well and report the results.

2. How is the PS approach adapted to deal with non-binary decision variables for the experiments?

---

> ### Author Response · Authors · 2024-11-25
> **Response to Reviewer u68k**
>
> ## Response to the weakness 1 and question 1
> You provided an excellent suggestion! We adopted your advice by using the initial solution as a hint for the 'completesol' heuristic method during the presolve phase, effectively employing the base solution as a warm start. We are pleased to discover that the performance of our VP-OR method has been further improved by integrating this warm-start strategy into our approach. Additionally, we added SCIP combined with the warm-start strategy as a baseline, with the results presented in Appendix C.7.
>
> **Detailed Results:** We observed that both SCIP and our VP-OR approach have improvements in the quality of feasible solutions under these conditions. The results are shown in the table below, where "NA" indicates cases where the method could not find a feasible solution within the time limit (T=10s).
>
> |  | bnd_1  | bnd_2 | bnd_3| mat_1|obj_1|obj_2|rhs_1|rhs_2|rhs_3|
> |-------|-------|-------|------|----|----|----|----|----|-----|
> |SCIP|0.16|NA|NA|0.23|0.00|0.39|0.50|0.00|0.00|
> | SCIP+completesol | 0.10|NA|NA|0.22|0.00|0.12|0.50|0.00|0.00|
> |VP-OR(Ours)|0.02|0.11|0.08|0.16|0.01|0.74|0.00|0.00|0.00|
> |VP-OR+completesol(Ours)|**0.02**|**0.11**|**0.06**|**0.16**|**0.00**|**0.06**|**0.00**|**0.00**|**0.00**|
>
> ## Response to the weakness 2
> We revised the description as follows: "VP-OR is more suitable for scenarios that require rapidly obtaining high-quality solutions in the short term. It converges quickly to find high-quality feasible solutions in the early stages of solving, but in the global scope, we also found that our method may encounter the possibility of getting stuck at suboptimal solutions."
>
> ## Response to the weakness 3 and question 2
> First, we would like to briefly explain the idea behind the LNS-based method proposed by Han et al [1]. Their approach focuses on predicting binary variables and then selects the top a% of binary variables that are predicted to be closest to 0 or 1, forming a set called $X_{select}$. For this set, a limit based on a distance $\delta$ is added. Specifically, they generate a new subproblem and solve it using the solver by **adding a constraint to the original MILP problem**:
> $\sum_{x\in X_{select}} |x-round(x_{pre})|<=\delta$, where $x_{pre}$ represents the probability of variable $x$ being predicted as 1, and round denotes the rounding function.
>
> As a result, even when general integer and continuous variables are present, **they are treated similarly to the non-fixed binary variables**. Therefore, we do not need to make additional adjustments for these variables. Instead, we rely on SCIP's existing capabilities to handle non-binary variables, which allows us to directly test the PS method on general MILP problems.
>
> [1] Qingyu Han et al. A gnn-guided predict-and-search framework for mixed-integer linear programming. ICLR, 2023.
>
> ## Response to the weakness 4
> We changed the use of "x%" to "a%" to avoid confusion with the notation used for decision variables.
>
> ## Response to the weakness 5
> We adjusted the size of Fig. 2, including increasing the font sizes of the legends and axis labels, so that it can be easily and clearly read.
>
> ## Response to the weakness 6
> We updated the formatting of the appendix.

---

> > ### Comment · Reviewer_u68k · 2024-11-26
> >
> > Thank you for responses which address my concerns. In particular, I appreciate you detailed comments and that you conducted additional experiments involving warm-starting - happy that my comments even helped you strengthening your results.
> >
> > I stand by my positive assessment of the paper.

---

### Official Review · Reviewer_ovGo · 2024-11-05

**Soundness:** 3
**Presentation:** 3
**Contribution:** 3
**Rating:** 6
**Confidence:** 3

**Summary:**

The paper introduces a re-optimization framework, VP-OR, for Mixed Integer Linear Programming (MILP) problems with dynamic parameters. The authors address real-world MILP applications, such as logistics and scheduling, where updates to parameters require quick re-optimization without starting from scratch. VP-OR combines a Graph Neural Network (GNN) model, which predicts variable probabilities, with an iterative Thompson Sampling approach to refine solutions by updating variable ranges and fixing values iteratively. The approach is tested on nine datasets, demonstrating its ability to find high-quality solutions faster than existing methods, including SCIP and other machine learning-based re-optimization techniques.

**Strengths:**

Originality: The paper presents an innovative approach to re-optimization for MILP problems with dynamic parameters, an area with limited solutions, especially for general MILP cases. The proposed VP-OR framework creatively combines machine learning through a Graph Neural Network (GNN) with probabilistic methods using Thompson Sampling.

Quality: The methodology is well-founded and executed. The use of GNNs for variable prediction is appropriately adapted to handle both binary and continuous variables in the MILP context, showing a solid understanding of the unique challenges in this domain.

Clarity: The paper is well-structured and logically progresses from problem formulation to methodology and results. Each component of the VP-OR framework, including variable prediction, Thompson Sampling, and iterative refinement, is explained in detail, making it easy for readers to follow the complex methodology.

Significance: VP-OR addresses a significant challenge in reoptimization for dynamic MILP problems, which are prevalent in real-world applications like logistics, production planning, and scheduling. By focusing on quick, high-quality reoptimization, the paper addresses a critical need for efficient solutions in time-sensitive scenarios.

**Weaknesses:**

Assumptions and Limitations in Variable Fixing: The VP-OR framework relies on predictions of feasible intervals for variables, which are used to fix values iteratively. However, if predictions are inaccurate, this approach can lead to suboptimal solutions or even infeasible ones.

Lack of Exploration on Convergence Guarantees: The paper does not provide formal convergence guarantees for the iterative refinement process, relying instead on empirical performance.

Scalability Analysis: The paper does not provide a detailed scalability analysis regarding the computational complexity of VP-OR as problem sizes increase.

**Questions:**

Are there theoretical or empirical convergence guarantees for VP-OR?

---

> ### Author Response · Authors · 2024-11-25
> **Response to Reviewer ovGo (Part 1/2)**
>
> ## Response to the weakness 1
> **Analysis of the prediction accuracy:** You have raised a valid point regarding the challenges associated with the variable fixing strategy, particularly when the predictions are inaccurate. This is indeed a common challenge in variable fixing approaches. However, it is important to highlight that in a reoptimization scenario, this challenge is less pronounced. This is because we leverage historical solution instances and intermediate solving process information(e.g. basis variables at each node’s LP relaxation) to aid our predictions. This approach significantly improves the accuracy of binary variable predictions compared to traditional end-to-end solving methods, which rely solely on modeling the problem as a bipartite graph and optimal solution values. Moreover, we can also provide bounds for integer and continuous variables, which are typically more challenging to handle.
>
> The table below demonstrates the predictive performance of both traditional Graph Neural Networks (GNN) and our approach in a reoptimization context(Re_GNN):
>
> |  | bnd_1  | bnd_2 | bnd_3 |
> |-------|-------|-------|-------|
> | Total binary var. |  1457.0 | 1457.0  | 1457.0  |
> |mispredicted binary var. (GNN) |  163.0 | 45.1 | 42.2 |
> |mispredicted binary var. (Re_GNN)|  **8.2** |**6.7**  | **4.5**  |
> | Total integer var. |  124.0 | 0.0  | 0.0  |
> |mispredicted integer var. (GNN) |  33.4 |0.0  | 0.0 |
> |mispredicted integer var. (Re_GNN)|  **17.4** |0.0  | 0.0 |
> | Total continuous var. |  0.0 | 301.0 | 301.0 |
> |mispredicted continuous var. (GNN) |  0.0 |  140.2  |  121.0  |
> | mispredicted continuous var. (Re_GNN)|  **0.0** |**0.0**  | **2.0** |
>
> Results show that by incorporating historical solution process information, Re_GNN achieves more accurate variable predictions compared to traditional GNN methods.
>
> **Our methods:** In our paper, specifically in Section 4.2, we address the challenge of **handling inaccurate variable predictions** by employing Thompson sampling in a multi-round solving strategy. For binary variables, we choose to fix them at 0 or 1, while for integer and continuous variables, we decide whether to apply the predicted bounds. The objective is to find an appropriate fixing strategy for each variable, adjusting it based on the results from each round.
>
> We adjust our fixing strategy using the Beta distribution parameters, $\alpha$ and $\beta$. The mean of the $Beta(\alpha,\beta)$ distribution is $\frac{\alpha}{\alpha + \beta}$. As these parameters increase, the distribution becomes more concentrated around the mean. With a prior of $Beta(\alpha,\beta)$, the posterior updates to $Beta(\alpha+1,\beta)$ or $Beta(\alpha,\beta+1)$. In each iteration, we fix a percentage a% of the variables. When we find a better solution, we update the Beta distribution for the remaining 1−a% of unfixed variables based on this new solution. We also compare the current strategy to the one from the previous round that gave the best solution. If a variable was fixed before but left unfixed in the current iteration, it indicates the previous strategy limited the solution quality. We update the Beta distributions for these variables to reflect this. In the next round, we resample the fixing strategy using these updated Beta distributions. Additionally, if the solution is infeasible, we implement a relaxation strategy by reducing the percentage of variables being fixed. This approach helps ensure that we can find feasible solutions by allowing greater flexibility in variable selection.

---

> > ### Author Response · Authors · 2024-11-25
> > **Response to Reviewer ovGo (Part 2/2)**
> >
> > ## Response to the weakness 2 and question 1
> > **Theoretical analysis**: For each variable, our goal is to decide whether to fix it at 0 or 1. Thompson sampling(TS) serves as a simple and elegant heuristic strategy. Based on the analysis by Russo et al. [1], the expected cumulative regret of TS up to time T is bounded by $O(\sqrt{log(|A|)dT})$, where $d$ represents the number of fixed variables, and $|A|=2^{C_n^d}$. TS strives to make decisions that improve the current feasible solution and explore arms that are not frequently selected.
> >
> > **Intuitive rationale for Thompson Sampling's effectiveness:** We offer insights into our intuitive rationale.
> > * When a variable is fixed frequently, the sum $\alpha+\beta$ in its Beta distribution becomes large, making the distribution narrow. If the variable's value has little impact on solution quality, it should not be fixed, and its expected value will be near the distribution center, away from 0 or 1. Conversely, if its value has a significant impact, the distribution's center will be close to 0 or 1, clearly indicating whether the variable should be fixed at one of these values.
> > * If a variable is not frequently selected, we cannot determine an appropriate decision for it confidently. In such cases, $\alpha+\beta$ remains small, creating a broader distribution. This allows for random numbers close to the edges (0 or 1), thus promoting exploration.
> >
> > [1] Russo D, Van Roy B. An information-theoretic analysis of thompson sampling[J]. Journal of Machine Learning Research, 2016, 17(68): 1-30.
> >
> > ## Response to the weakness 3
> > **Computational complexity analysis:** The primary computational complexity of VP-OR arises from the Thompson Sampling process. In the sampling phase of Thompson Sampling, we sample the probability p for binary variables and select a certain percentage (a%) of variables based on the value of $min(p, 1 - p)$ by sorting them. For integer and continuous variables, we sample to determine whether they should be fixed and select the top a% of variables based on this criterion. This step has a time complexity of **O(nlogn)**, where n is the number of variables. In the parameter update phase of Thompson Sampling, we update the parameters for each variable once. This step has a time complexity of **O(n)**.
> >
> > **Experimental Results:** We tested the sampling time and parameter update time for each instance, and the average values for each dataset are presented in the following table:
> > |  | bnd_1  | bnd_2 | bnd_3| mat_1|obj_1|obj_2|rhs_1|rhs_2|rhs_3|
> > |-------|-------|-------|------|----|----|----|----|----|-----|
> > |Var. num|3117|1758|1758|802|360|745|12760|1000|1000|
> > | Sampling Time(s) |0.008|0.005| 0.003|0.013|0.002|0.016|0.102|0.002|0.002|
> > | Parameter update Time(s) | 0.002 |0.001 |0.000|0.009|0.001|0.010|0.070|0.001|0.001|
> >
> > The results show that the time spent on distribution sampling and parameter updates is significantly less than the time required by the solver to find solutions.

---

> > > ### Author Response · Authors · 2024-11-27
> > >
> > > Dear Reviewer ovGo,
> > >
> > > With the rebuttal deadline approaching, we're eager to hear your thoughts on our response and revisions. Have we adequately addressed your concerns? Your feedback is crucial to us, and we're committed to further improving our paper based on your insights.
> > >
> > > Thank you for your time.
> > >
> > > Best regards,
> > >
> > > Authors

---

### Author Response · Authors · 2024-12-02
**General Response**

We thank all the reviewers for their thoughtful feedback and their time.

We are encouraged to see that they found the ideas in our paper innovative (Rev. ovGo), to form original contributions (Rev. u68k) and found the approach well-founded (Rev. ovGo) and found the experiments to be fairly strong (Rev. u68k) and to be useful for future works (Rev. u68k). We especially thank Reviewer u68k for inspiring us to incorporate solutions from previous instances as a hint to enhance the heuristics in the pre-solving process, which helps us further improve our methods.

In this work, we address dynamic environments that require **rapid responses** to **minor parameter changes**. We found that most integer and continuous variables cannot be accurately predicted if relying solely on optimal solution values and modeling the problem as a bipartite graph in Table 11. To overcome this, we propose a method to predict the **bounds** for these variables, leveraging **historical** branch-and-bound processes from previously solved instances. This significantly improves the accuracy of variable predictions. To meet **time-critical** demands, we adopted a fixing strategy. This approach results in fairly short solving times for each iteration compared to traditional LNS strategies (at least 10 times shorter, as shown in Table 2). Consequently, we employ Thompson sampling to **iteratively select** the set of variables to fix, based on solutions obtained in each iteration.



Reviewer au9E mentioned that predicting ranges for integer and continuous variables in our paper lacks novelty, identifying it as the primary weakness of the work. We respectfully disagree. They provide one citation [1] which is already appropriately cited in our paper. In our individual responses, we have provided **experimental results** showing that the cited method cannot be directly applied to our reoptimization dataset. We have discussed the **key insights** of our paper in detail, and we believe these insights are not found in prior works.


Reviewer 6YTp mentioned the datasets we selected as the main weakness. The datasets we tested are based on the **MIP Workshop 2023 Computational Competition on Reoptimization** [2]. These datasets primarily originate from **MIPLIB** [3] and some **real-world time-sensitive** reoptimization problems. MIPLIB is a well-known and widely used collection of benchmark problems in the field of mixed-integer linear programming (MILP). It is frequently maintained and updated to include a diverse set of test problems sourced from various **real-world applications and industries**. They are highly relevant to real-world reoptimization scenarios.

**References:**

[1] Nair V, Bartunov S, Gimeno F, et al. Solving mixed integer programs using neural networks[J]. arXiv preprint arXiv:2012.13349, 2020.

[2] Bolusani S, Besançon M, Gleixner A, et al. The MIP Workshop 2023 Computational Competition on Reoptimization[J]. Mathematical Programming Computation, 2024: 1-12.

[3] Gleixner A, Hendel G, Gamrath G, et al. MIPLIB 2017: Data-driven compilation of the 6th mixed-integer programming library[J]. Mathematical Programming Computation, 2021, 13(3): 443-490.

---

> ### Author Response · Authors · 2024-12-02
> **Experiments and changes based on reviewers' comments**
>
> Based on the reviewers’ comments, we updated the draft to include new experiments, discussion and some other minor changes. Here is the summary of the changes.
>
> **Primary Changes:** Experiments and Results
> * **Historical Information Ablation Study (Rev. ovGo)**: We present the comparison results between the reuse of historical solving information and the traditional vanilla bipartite graph predictions. TLDR: The reuse of historical solving information significantly improves the accuracy of variable predictions. Described in Section C.5.
> * **Computational Complexity Analysis (Rev. ovGo)**: We included a complexity analysis of the Thompson Sampling process and tested both the sampling time and parameter update time. Described in Section C.6.
> * **Comparison with SCIP Using Historical Solutions (Rev. u68k)**: We added SCIP combined with the warm-start strategy as a baseline. TLDR: Both SCIP and our VP-OR approach have improvements in the quality of feasible solutions with the warm-start strategy. Described in Section C.7.
> * **Larger Sample Size Experiment (Rev. au9E)**: We tested the performance of the methods with more examples. TLDR: The results on the expanded test set are consistent with our previous tests. Described in Section C.8.
> * **Additional Baseline (Rev. au9E)**: We presented the results of an end-to-end method, Light-MILPopt, for comparison. TLDR: Light-MILPopt's performance on reoptimization datasets is slightly better than but quite close to the end-to-end baseline we used for comparison, PS. Described in Section C.9.
> * **Re_Tuning Configuration (Rev. 6YTp)**: We found that Re_Tuning may disable presolving and cutting plane modules, which can adversely affect its performance under strict time constraints. We tested Re_Tuning with these modules enabled, observing notable improvements in its early-stage performance. Detailed explanations are provided in Section C.2.
>
> **Secondary Changes:** Minor/Writing
> * Minor clarifications and formatting adjustments based on comments from Rev. u68k and Rev. au9E.
>
> Once again, we genuinely appreciate reviewers' dedication to the review process.

---

### Meta-Review · Area_Chair_uvpE · 2024-12-22

**Metareview:**

This paper addresses the low effectiveness of the standard re-optimization techniques in Mixed Integer Linear Programming.  To overcome this issue, the novelty of this paper lies in proposing a two-stage reoptimization framework that uses bandit-type algorithms to fix part of variables and predict the rest of them. However, the main concerns from the reviewers lie in the lack of novelty compared to some recent works, and the limited experiments to demonstrate the power of this framework. I read the main paper and the reviews, and agree with some of the concerns.

**Additional Comments On Reviewer Discussion:**

One negative reviewer was not engaging during the discussion. I have read the review and the response from the author.

---

### Decision · Program_Chairs · 2025-01-22

Reject